# Index establishment and capability evaluation of space-air-ground remote sensing cooperation in geohazard emergency response

Yahong Liu[1], Jin Zhang[2]

[1] Department of Earth Science and Engineering, Taiyuan University of Technology, Taiyuan 030024, China;

[2] Department of Surveying and Mapping Science and Technology, Taiyuan University of Technology, Taiyuan 030024, China

*Correspondence to*: Jin Zhang (zjgps@163.com)

**Abstract.** Geohazard emergency response is a disaster event management act that is multifactorial, time critical, task intensive and socially significant. To improve the rationalization and standardization of space-air-ground remote sensing

collaborative observations in geohazard emergency responses, this paper comprehensively analyzes the technical resources of remote sensors and emergency service systems and establishes a database of technical and service evaluation indexes using MySQL. Based on the database, we propose the method of using technique for order preference by similarity to an ideal solution (TOPSIS) and a Bayesian network to evaluate the synergistic observation effectiveness and service capability of remote sensing technology in geohazard emergency response, respectively. We demonstrate through experiments that

using this evaluation can effectively grasp the operation and task completion of remote sensing cooperative technology in geohazard emergency response. This provides a decision basis for the synergistic planning work of heterogeneous sensors in geohazard emergency response.

**Keywords:** Geohazard; Remote sensing cooperation; Index database; Capacity evaluation

## 1 Introduction

Geohazards are earthquakes, mountain collapses, landslides, debris flows, ground collapses, ground fissures, land subsidence and other hazards related to geological processes that endanger people's lives and property and are caused by natural factors or human activities. According to the United Nations Office for Disaster Risk Reduction (UNDRR), the human casualties caused by geological hazards since 1990 have been concentrated in the Asia-Pacific region and Africa for a long time, with 2010-2019 the decade with the highest economic losses caused by disasters (UNDRR Annual Report, 2019; UNDRR GAR

2019). To respond to sudden geological hazards and mitigate damage, it is necessary to carry out hazard emergency response quickly after the occurrence of a hazard, provide emergency assistance for victims and seek to stabilize the situation and reduce the probability of secondary damage (Johnson, 2000).

Earth observation technology provides key technical support during geohazard emergency response (Butler et al., 2005). With the development of global earth observation technology, the performance of remote sensing technology is constantly

improving, the number of sensors continues to increase and a multiplatform observation system for satellites, aerials, unmanned aerial systems (UASs)and the ground has gradually been established (Toth et al., 2016). There are many online resources for recording remote sensing information, and the NASA master directory (NSSDC, 2020) provides a mechanism for retrieving satellite names, classifications or launch dates to obtain descriptions of relevant satellite information and data collection. The CEOS Missions, Instruments, and The Measurements (MIM) Database is divided into Agencies, Missions, Instruments, Measurement and Datasets modules with a focus on current and future satellites, sensors and measurement capabilities (CEOS, 2020). The Observing Systems Capability Analysis and Review tool (OSCAR, 2020) database is divided into a description of information about the satellite and its sensors and a sensor capability assessment analysis.

At present, most of the Earth observation technology resources operate independently, and when faced with specific geohazard emergency response tasks, the space-air-ground remote sensor resources show both "many" and "few." That is, although sensor resources are abundant, it is difficult to find suitable and available sensors quickly, and this affects the efficiency of observing mission responses. The main reason is that remote sensing systems of various types are very different in terms of observation modes, applications and processing methods. In addition, resources are deployed in a distributed fashion, are described in their own independent formats, lack correlation mechanisms and cannot be detected in a timely manner (Li et al., 2012). To improve the efficiency of emergency response, a number of organizations and mechanisms have been established internationally to synergize these resources, including the Committee on Earth Observation Satellites (CEOS), Integrated Global Observing Strategy (IGOS), International Charter Space and Major Disasters (CHARTER) and United Nations International Strategy for Disaster Reduction (UNISDR). The International Strategy for Disaster Reduction (UNISDR), United Nations Platform for Space-based Information for Disaster Management and Emergency Response (UN-SPIDER), Disaster Monitoring Constellation (DMC) and Copernicus EMS are mainly oriented to international major disaster emergency responses such as the Wenchuan earthquake (PAN et al., 2010), Haiti earthquake (Duda et al., 2011) and Japan earthquake (Kaku et al., 2015). In addition to establishing collaborative emergency response with satellite remote sensing, in the face of the diversified needs of actual geohazard emergency response, collaboration between satellites and other multiple remote sensing platforms has become an important development direction of remote sensing technology (Li et al., 2017). This is characterized by the ability to integrate the observation advantages of each platform to effectively shorten the observation time, expand the coverage and improve the accuracy of observation data (Asner et al., 2012; Nagai et al., 2009). Haghighi et al. (2019) used multi-SAR satellite sensors for the analysis of spatial and temporal processes of ground subsidence in the Iranian region of Drangheh, Hermle et al. (2021) used to verify the feasibility of optical remote sensing in landslide hazard warning through a combination of high-resolution satellite and UAV data and Lu et al. (2019) mapped landslide inventories based on multi-remote sensing sensor data, Ventisette et al. (2015) described data acquisition using satellite and ground-based sensors in landslide disaster response, and Huang et al. (2017) proposed a complete set of methods for geohazard emergency investigation using UASs. In these remote sensing collaborative disaster emergency applications, by linking different types of remote sensors and coupling them to form an independent and dynamically adaptable and configurable space-air-ground remote sensing collaborative observation system, the complementary

advantages of remote sensing observation platforms are brought into full play. However, there is no sensor discovery process in these studies, and there is a lack of selection criteria and capability evaluation of sensors in different collaborative applications.

The observation tasks under geohazard emergencies are complex and diverse and have certain requirements in terms of timeliness and accuracy, and it is especially important for decision-makers to make comprehensive discoveries and to establish accurate collaborative planning and rapid scheduling of massive sensors in a specific emergency response situation. How to quickly and rationally arrange the sensors that meet the geohazard emergency response needs in the sensor web environment to optimize resource utilization is the key issue in remote sensing collaborative observation. This work focuses on establishing a link between geohazard emergency response events and sensors, constructing indicators for evaluating the technical capabilities of sensors and evaluating geohazard emergency service capabilities. Wang Wei et al. (2013) proposed a mission-oriented assessment of the observational capabilities of imaging satellite sensor applications with the horizontal resolution, revisit period and observation error as indicators. Hong Fan et al. (2015) proposed a sensor capability representation model to describe typical remote sensor capabilities for soil moisture detection applications. Zhang Siyue et al. (2019) proposed a model for evaluating the effectiveness of observations and data downlinks for low-orbiting satellites. Hu et al. (2019) constructed the observation capability information association model (OCIAM) for the selection of sensors and their combinations, and further proposed the sensor observation capability object field (SOCO-Field) to construct sensor associations for a specific emergent geographical environment observation task (GeoTask) (Hu, 2020), and Wang et al. (2020) introduced the space-ground maximal coverage model with multiple parameters (SGMC-MP) to complete sensor mission planning. The current research data on remote sensor capabilities are relatively scarce and focus on evaluating the inherent capabilities of individual satellite remote sensors with a single object of evaluation, making it difficult to meet the needs of multisensor and multigeohazard emergency response tasks. Thus, it is necessary and timely to establish collaborative observation capability indexes for space-air-ground remote sensor resources and to conduct evaluations of geohazard emergency response service capabilities.

## 2 Data

Given the richness of remote sensor technology resources, the service system for emergency response to geohazards has been improved in application practice. An important question how to fully discover and use the existing sensor technology to meet the target observation needs and achieve the optimal effect of resource utilization for different application services. To allocate sensor resources scientifically, improve the rationality and effectiveness of cooperative observation and obtain the required information to a greater extent, this section establishes an index database for comprehensive analysis of the technical performance and emergency service system of space-air-ground remote sensing, and realizes the integrated management of the technical performance data of various types of remote sensors and emergency service information.

## 2.1 Sensor technology resource emergency service system

Current remote sensors can be divided into satellite, aerial and terrestrial types according to the platforms on which they are mounted (Grün, 2008). Satellite remote sensing is divided into land satellites, meteorological satellites and ocean satellites according to their fields of operation. Land satellites are mainly used to detect the resources and environment on the earth's surface and contain a variety of sensor types such as panchromatic, multispectral, hyperspectral, infrared, synthetic aperture radar, video and luminescence (Belward et al., 2015). Meteorological satellites observe the earth and its atmosphere, and their operations can be divided into Sun-synchronous polar orbit and geosynchronous orbit (NSMC, 2020; Wang et al., 2018). Oceanic satellites are dedicated satellites that detect oceanic elements and the marine environment with optical payloads generally including watercolor water thermometers and coastal zone imagers and microwave payloads including scatterometers, radiometers, altimeters and SAR (Fu et al., 2019). The countries and regions in the world that currently have autonomous remote sensing satellites include the United States, France, ESA, Germany, Israel, Canada, Russia, China, Japan, Korea and India. The main satellite launches are shown in Table A1. Aerial remote sensing is a technology that uses aircraft, airships and UVAs as sensor carriers for detection (Colomina et al., 2014). Different airborne remote sensing devices have been developed to face various remote sensing tasks. These devices include digital aerial cameras, LiDAR, digital cameras, imaging spectrometers, infrared sensors and min SAR. Ground remote sensing systems have two states: mobile and static. A mobile measurement system executes rapid movement measurement by means of vehicles (e.g., cars and boats) and consist of sensors such as CCD cameras, cameras, laser scanners, GPS and inertial navigation systems (INSs) (Li et al., 2015). These can acquire the geospatial position of the target while collecting realistic images of the features. Static state measurement refers to the installation of sensors in a fixed place and includes laser scanners, cameras, ground-based SAR and surveying robots. These can form a ground sensor web through computer network communication and geographic information service technology.

In the face of geohazard emergency responses, space-air-ground remote sensors establish associations through collaborative planning to form a collaborative observation service system based on the process of "observation-transmission-processing-distribution," as shown in Fig. 1. In the event of a geological disaster, the emergency command center responds quickly, planning observation missions according to observation needs and the current technical environment(①,②). After remote sensing systems carry out observation missions(③), the data is received, processed and distributed through the data center, providing emergency services mainly based on geographic information(④,⑤,⑥).

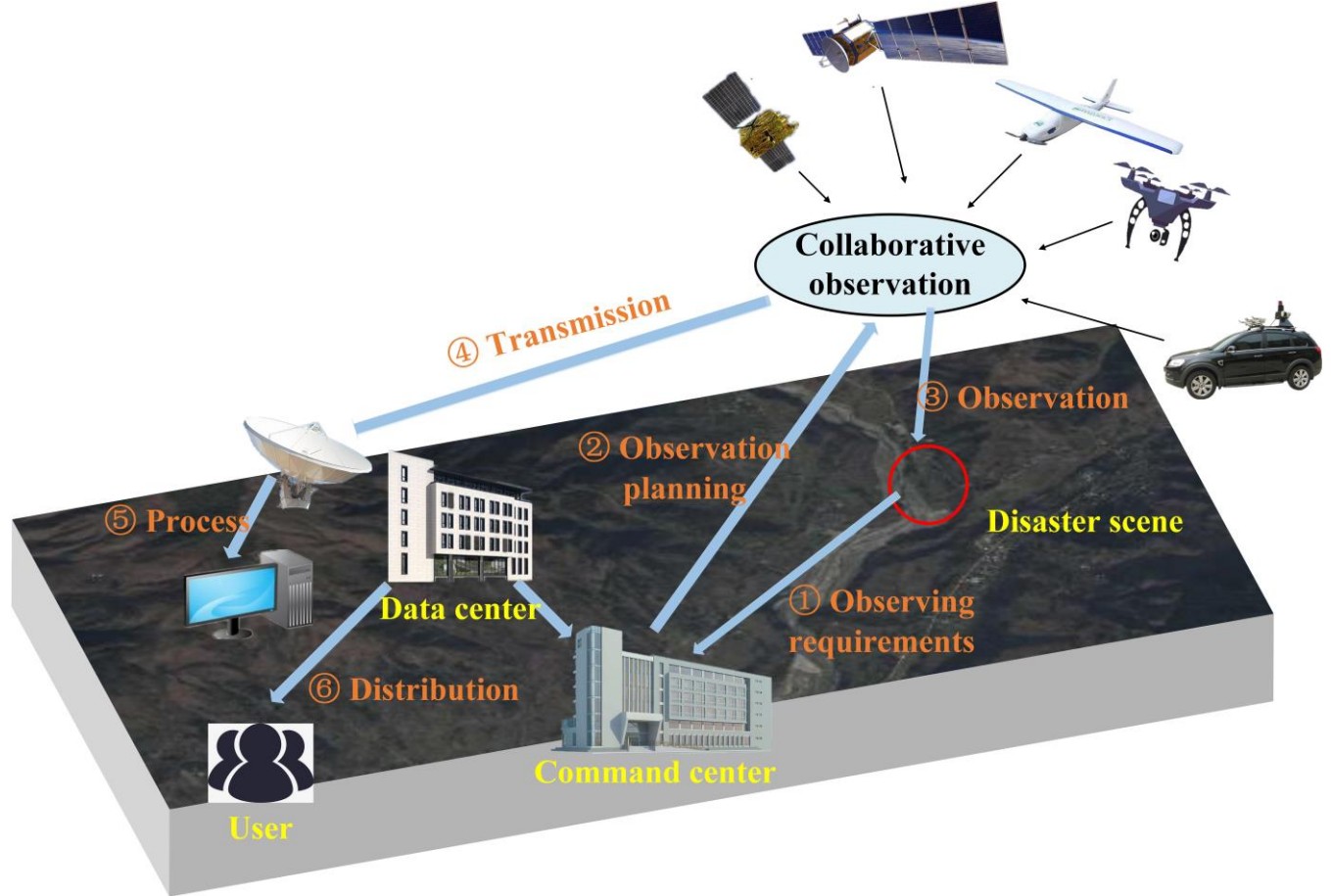

**Figure 1. Collaborative remote sensing observation service system for geohazard emergency response**

The geographic information services provided by the remote sensing emergency service system are shown in Fig. 2. These services include data processing, data products, data services, model services, functional services and warning services. Data processing refers to the process and method of obtaining effective emergency information from the collected data and includes the data processing method, feature extraction, image classification and image analysis. Data products refer to the quality and current potential of various types of remote sensing products. Data services provide disaster-related basic data, thematic data and analysis data through Web Map Service (WMS), Web Feature Service (WFS),Web Coverage Service (WCS) and Web Map Tile Service (WMTS). Functional services provide quantitative, qualitative, characterization and visualization of geospatial phenomena through spatial analysis services, terrain analysis services and visualization services. Model services provide various models for calculation, analysis, anomaly identification, damage assessment, situational assessment, evaluation, decision-making and optimization. Warning services provide early warning of disasters with regard to space, time and situation.

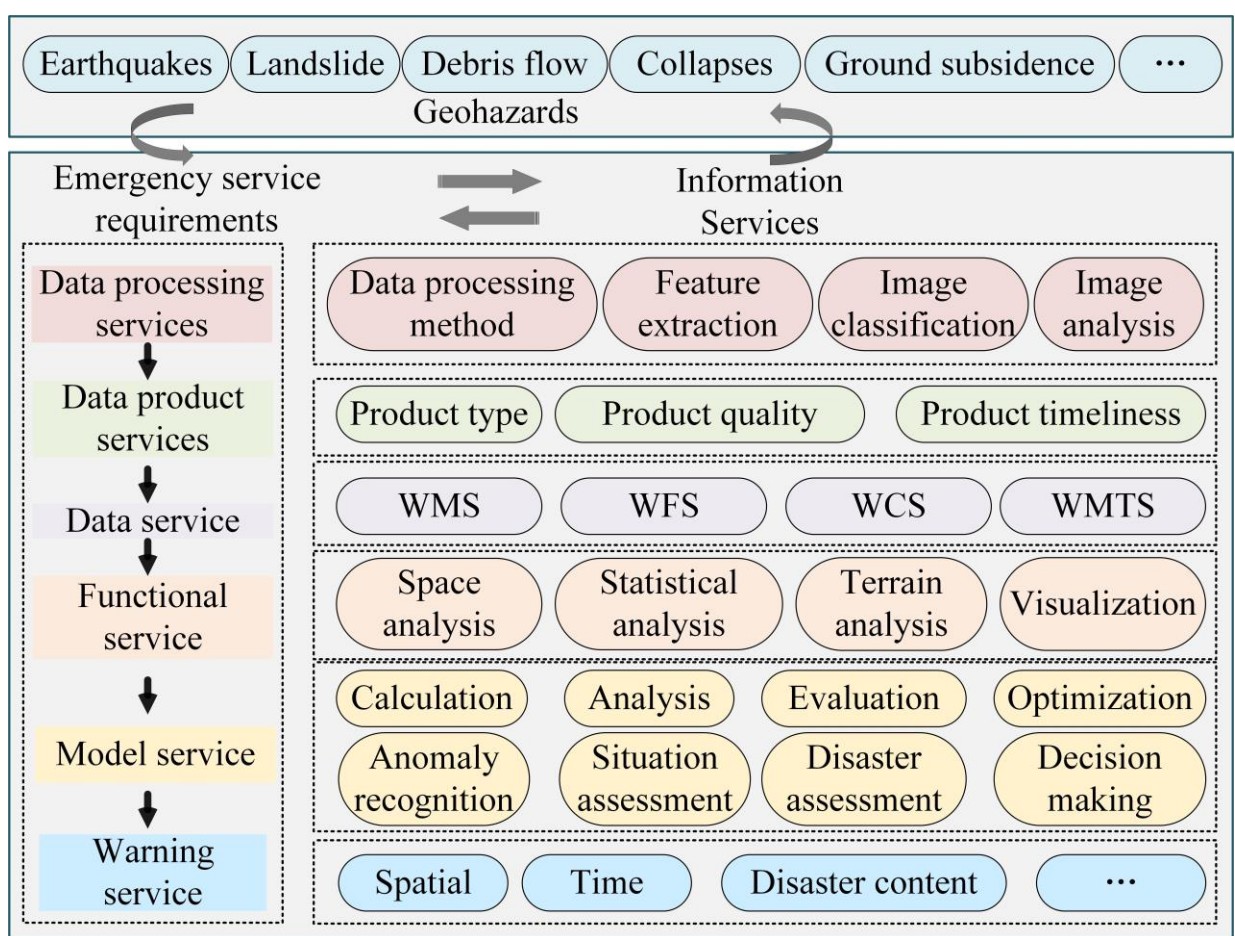

135

**Figure 2. Emergency geographic information service**

## 2.2 Space-air-ground remote sensing index database

The existing database focuses on satellite remote sensing resources and does not form a unified management for aerial, UAV, ground remote sensing platforms or their sensor information when facing the demand of remote sensing cooperative

140 response in actual disaster emergencies. This paper establishes an integrated space-air-ground remote sensing index database covering satellite, aerial and ground platforms that adopts MySQL for storage management. MySQL is an open-source relational database management system that supports multiple storage engines such as MyISAM and InnoDB. It also supports spatial data objects in terms of geographic information by complying with the OpenGIS Geometry Model of the Open Geospatial Consortium (OGC), provides various Application Programming Interfaces (APIs) and supports multiple

145 operating systems and development languages. Thus, MySQL can provide good Web service applications. The database is divided into two parts: SAT_RS, the sensor technical performance index database; and SE_RS, the emergency service evaluation index database.

The technical performance indicators of sensors in SAT_RS are their various capability characteristics under normal operation as reflected by technical parameters. The indicators are independent between different types of sensors. The parameters vary, and the technical indicators are also diverse. In the face of complex geohazard emergency response needs, how to select the appropriate sensors to accomplish the observation tasks requires the classification of existing sensors according to their capabilities and a synthesis of technical indicators. In this regard, this study collected and summarized the technical parameters of various types of sensors, referred to the selection of indicators in satellite online data repositories (NSSDC 2020, CEOS 2020 and OSCAR 2020) and the experience of relevant professionals in using them, analyzed the information of various types of sensors and established a more complete sensor technology index system. This is shown in Table 1 below.

The indicators in the table are mainly considered in terms of the amount of information, timeliness, validity (accuracy) and expressiveness of data acquisition. The indicators are selected for different types of sensor technical indicators. The amount of information is used to eliminate the uncertainty in the expression of spatio-temporal characteristic information in the observed data, including the temporal extent, the spatial area, and the degree of spatial details such as geometry and attributes, reflecting the intensity of the acquired information, and is related to the breadth and depth of the sensor's role with regard to the scan width, side-swing capability, measurement range, etc. Timeliness refers to the self-conscious dynamism of the sensor system and the degree of sensitivity and response to the task, and is related to the responsiveness and execution efficiency of the sensor. Factors include the revisit cycle of the satellite, preparation time of the UAV and endurance. Validity expresses the accuracy of the acquired information with regard to resolution, quantization level and measurement accuracy, for example. Expressiveness describes the representational form of the information. Note that the same indicator has multiple effects on data acquisition, such as the spatial resolution of the satellite having an impact on both the amount and validity of information. Thus, it is necessary to set a comprehensive evaluation indicator of information acquisition capability in different dimensions.

**Table 1. Sensor technical indexes**

| Type | Technical indexes |
|------|-------------------|
| Optical satellite | Spatial resolution |
| | Spectral resolution |
| | Radiation resolution |
| | Revisit time |
| | Swinging ability |
| | Swath width |
| SAR satellite | Wave band |
| | Polarization |
| | Spatial resolution |

| | |
|---|---|
| | Revisit time |
| | Swath width |
| | Incidence angle |
| Photogrammetry | Resolution |
| | Data type |
| | Preparation time |
| LiDAR | Point cloud density |
| | Measuring range |
| | Measurement accuracy |
| UAV | Endurance time |
| | Cruising speed |
| | Payload |
| Mobile measurement | Measuring range |
| | Data type |

The emergency service indexes in SE_RS refer to the capacity evaluation indexes of the emergency service system associated with the event. The space-air-ground remote sensing geohazard emergency service capacity evaluation index system established is shown in Table 2 below. The index is measured in the three aspects of data acquisition, processing and information service. The specific content of the indexes should be determined in conjunction with the responding emergency event.

**Table 2. Indexes for evaluation of emergency service capacity**

| Constitute | Criterion |
|---|---|
| Data acquisition | Technology, Data Volume, Timeliness, Responsiveness |
| Data processing | Methodology, Speed, Quality |
| Information services | Demand, Quality, Timeliness |

The database design process is divided into information analysis, structure design, storage settings and data storage. By analyzing the massive sensor information, we set the attribute fields from the carrying platform, set the technical characteristics of each type of sensor and operation status, store the corresponding data in the database and establish unified management and finally design 20 kinds of tables. This is shown in Fig. 3. SAT_RS records the basic satellite, aerial and terrestrial information through three tables: RS_Satellite, Sensor_Aerial and RS_Terrestrial. Then, SAT_RS establishes a technical characteristics index table for different types of sensors on different platforms according to the technical index system shown in Table 1. This includes SatelliteSensor_Optical, SatelliteSensor_SAR, UAS, ImageSpectrometer, DigitalCamera, AirbronrLiDAR, MinSAR, MMS and 11 other types of tables. The RS_Task table in SE_RS links tasks among sensors and records the observation tasks they perform, including observation equipment and observation time. The

evaluation indexes in RS_DataProcessing and RS_Service are set according to the guidelines of Table 2 and the specific geohazard remote sensing emergency service events.

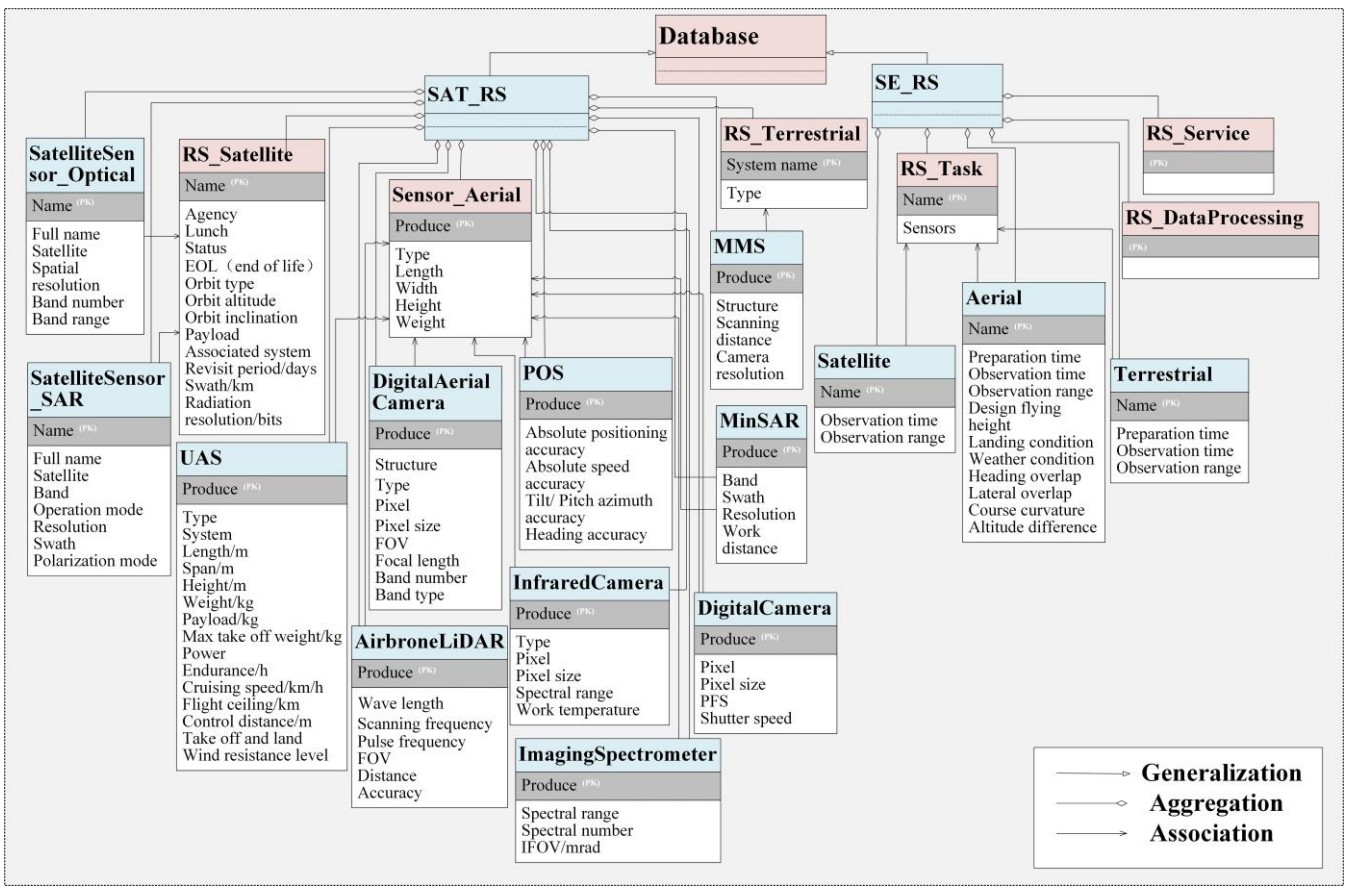

**Figure 3. Unified Modeling Language (UML) of database**

At present, the SAT_RS database records approximately 150 satellites and their corresponding sensor data from many countries and organizations including the United States, France, ESA, Russia, Japan, Korea, India and China; more than 100 commonly used aerial remote sensor product families; more than 50 UAV products; and dozens of ground mobile measurement systems. A partial display is shown in Fig. 4. The features of SAT-RS are as follows: (1) Wide data coverage, support for satellites, aviation platforms (including UAVs), terrestrial multiplatforms and multiple types of remote sensors.

(2) Indexing of sensor technical performance and support for evaluation calculations. (3) Data support for sensor ML descriptions.

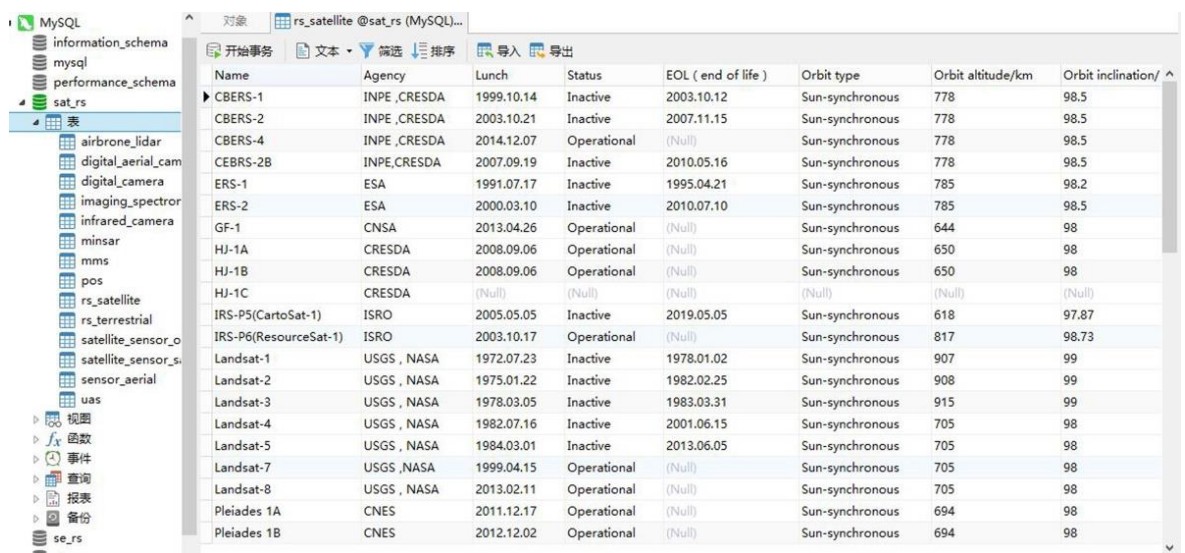

**Figure 4. SAT_RS database (partial)**

## 3 Methodology

The methods commonly used to evaluate the system capabilities are the analytic hierarchy process (AHP) (Emrouznejad et al., 2017), fuzzy integrated assessment (Kahraman et al., 2015), technique for order preference by similarity to an ideal solution (TOPSIS) (Zhang, 2015), rank sum ratio (RSR) (Tian, 2002) and Bayesian network (BN) (Heckerman, 2008), all of which have their own characteristics. The evaluation studied in this paper is a complex and flexible multisystem and multi-influencing factor problem. In order to improve the scientific nature of the evaluation and make full use of the advantages of

various methods, TOPSIS and Bayesian-network-based evaluation methods are used for remote sensing collaborative observation and service capability, respectively, while RSR is used to determine the weights in TOPSIS calculation. The evaluation process is shown in Figure 5 below.

TOPSIS can eliminate the influence of different indicator magnitudes and make full use of the information of the original data. This is a common method for multiobjective decision analysis of limited solutions in systems engineering. Since this

method has no strict restrictions on the distribution, quantity and magnitudes of evaluation data, it is flexible in application and can be well adapted to the changes of indicators involving many types of sensors. Meanwhile, the use of RSR to determine the weights combines the Score Ratio (SR) and empirical weights. This overcomes to some extent the subjectivity of determining the weights and makes the evaluation results more reflective of objective facts.

A Bayesian Network is a probabilistic graphical model based on the dependency relationships among variables, and the

evaluation of emergency response service capability using this method has the following advantages. First, the emergency response process can be divided into a number of coherent and causally related links such as data acquisition, processing, information extraction and forming an emergency service chain. The evaluation of this service capability with backward and

forward correlation is suitable for modeling with directed graphs. Second, there are uncertainties in each link of emergency services that are suitable for probabilistic methods.

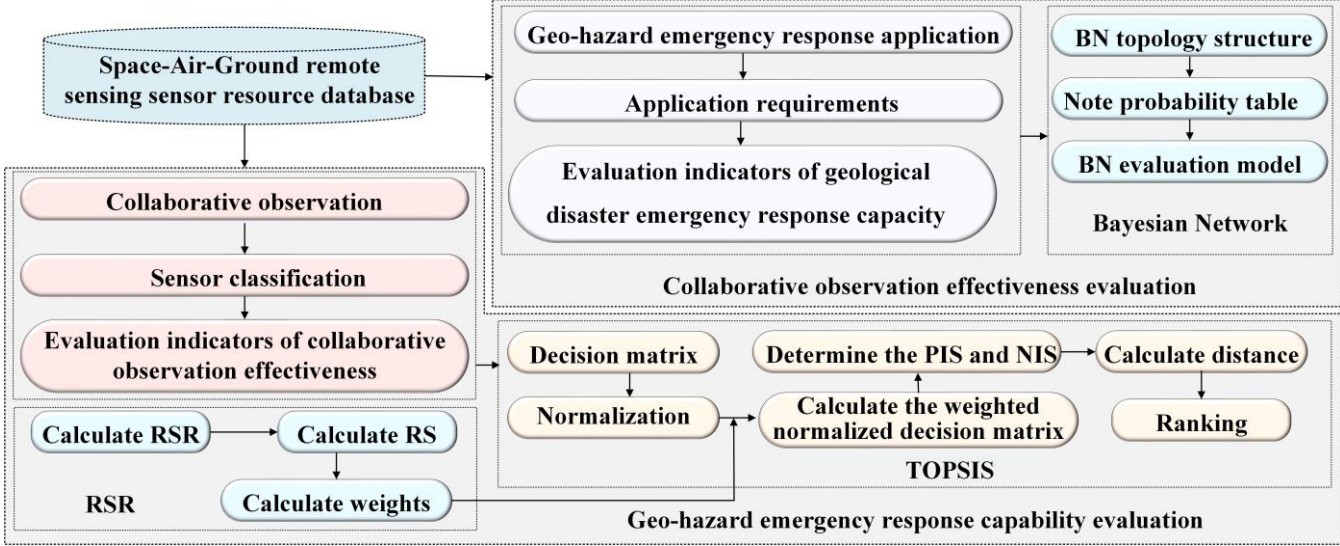

**Figure 5. Evaluation process**

### 3.1. TOPSIS

TOPSIS is commonly used in multiple-criteria decision-making (MCDM). The basic principle is to rank the evaluated objects by detecting the distance between the positive ideal solution (PIS) and the negative ideal solution (NIS). The evaluation object is best if it is closest to the PIS and farthest away from the NIS, where the PIS is composed of the best value of any alternative under the corresponding evaluation index. The NIS has the opposite logic. The evaluation process is as follows:

(1) Identify a decision matrix: Assuming that the evaluation object in an MCDM is composed of $m$ combinations of remote sensing cooperative work , $n$ evaluation indicators and the value of the $j$ th evaluation indicator for the $i$ th object is $a_{ij}$ ($1 \leq i \leq m, 1 \leq j \leq n$),the decision matrix $A = (a_{ij})_{m \times n}$ is as follows:

$$A = \begin{bmatrix} a_{11} & \cdots & a_{1n} \\ \vdots & \ddots & \vdots \\ a_{m1} & \cdots & a_{mn} \end{bmatrix} \tag{1}$$

(2) Indicators are treated with the same trend. To maintain the same direction of change for all indicators, a reciprocal method was used to convert negative indicators into positive indicators. The negative indicators refer to indicators with smaller values for better results, such as spatial resolution, revisit cycle, UAV preparation time, etc., vice versa for positive indicators, such as swath width, payload, etc.

(3) Normalize the decision matrix. The evaluation indicators have different attribute dimensions, and it is necessary to transform various attribute dimensions of the indicators into nondimensional attributes. The normalization decision matrix is $B = (b_{ij})_{m \times n}$, and the normalized value is computed as

$$b_{ij} = \frac{a_{ij}}{\sqrt{\sum\limits_{i=1}^{m} a_{ij}^2}} \tag{2}$$

(4) RSR determines weights. RSR is a statistical analysis method that combines the advantages of classical parametric estimation and modern nonparametric estimation. RSR refers to the average or weighted average of the rank totals of rows
(or columns) in a table and is based on the concept of converting indicator values into dimensionless statistical ranks and ratios by using statistical distribution, probability theory and regression analysis methods to evaluate and classify programs. For matrix $B$ obtained after the normalization process, the RSR is calculated as

$$RSR_j = \frac{\sum\limits_{i=1}^{m} R_{ij}}{m \bullet n} (1 \leq i \leq m, 1 \leq j \leq n) \tag{3}$$

$R_{ij}$ denotes the rank corresponding to the index value $b_{ij}$ of the $j$th evaluation index in the $i$th evaluation object, and the formula for determining the weight of each index using RSR is as follows:

$$W_j = \frac{SR_j \bullet W'_j}{\sum\limits_{j=1}^{n} SR_j \bullet W'_j} \tag{4}$$

The SR reflects the proportional relationship between the levels of each indicator and is calculated from RSR [Eq. (5)]. $W'$ is an empirical weighting factor.

$$SR_j = \frac{RSR_j}{\sum\limits_{j=1}^{n} RSR_j} \tag{5}$$

(5) Calculate the weighted normalized decision matrix. Multiplying the normalized processed matrix by the determined weight vector $W = [w_1 \cdots w_n]^T$ results in a weighted normalized decision matrix $C = (c_{ij})_{m \times n}$:

$$c_{ij} = b_{ij} \bullet w_j \tag{6}$$

(6) Computation of the PIS and NIS.

The positive ideal solution:

$$c_j^+ = \max c_{ij} \tag{7}$$

The negative ideal solution:

$$c_j^- = \min c_{ij} \tag{8}$$

PIS represents the indicator value of the most desirable synergistic solution inferred from $m$ combinations of approaches, and vice versa for NIS.

(7) Computation of distance each alternative from PIS and NIS:

Distance from the PIS is

$$d_j^+ = \sqrt{\sum_{i=1}^{m}(c_{ij} - c_j^+)^2} \tag{9}$$

Distance from the NIS is

$$d_j^- = \sqrt{\sum_{i=1}^{m}(c_{ij} - c_j^-)^2} \tag{10}$$

(8) Computation of relative closeness and ranking of alternatives:

The relative closeness is defined as

$$s_i = \frac{d_j^-}{d_j^+ + d_j^-} \tag{11}$$

The larger the $s_i$ value, the higher the ranking, the more desirable the remote sensing cooperative method.

## 3.2. Bayesian Network


The Bayesian Network, also known as a belief network, is a probabilistic graph model that was first proposed by Judea Pearl in 1985. The Bayesian Network applies probability theory to the reasoning of uncertainty problems, and its network topology is a directed acyclic graphical (DAG) with the ability to express and reason about uncertainty knowledge.

The nodes of a Bayesian Network represent random variables, and the directed links (edges) between the nodes indicate the

conditional dependencies between the random variables. All nodes pointing to node $M$ are called the parent nodes of $M$, $M$ is called the child node of its parent and variables without a parent node are called root node variables. All nodes have a corresponding node probability table (NPT) expressing the probability of occurrence of a random event, with the probability of the root node being the prior probability and the probability of the child node indicating all possible conditional probabilities of that node relative to its parent node (posteriori probability).

Fig. 5 illustrates a simple Bayesian Network where $A$, $B$, $C$ and $D$ are four variables, parent node $A$ is the root node, $B$ and $C$ are child nodes of $A$, and $D$ depends on variables $B$ and $C$. The joint probability distribution of the nodes is expressed as follows:

$$P(A,B,C,D) = P(A)P(B\mid A)P(C\mid A)P(D\mid B,C) \tag{12}$$

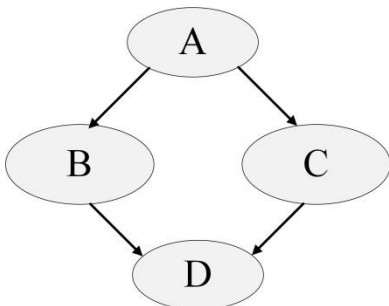

**Figure 6. Diagrammatic depiction of Bayesian model**

Probabilistic reasoning is one of the main uses of Bayesian Network. The reasoning essentially involves calculating a posteriori probabilities using conditional independence among random variables to calculate the a posteriori probability distribution of some other variables if the values of some variables in a Bayesian Network are known.

## 4 Results and Discussion

By managing indicators through the database, evaluation of the collaborative capability of space-air-ground remote sensing
technology in geohazard emergency response is established and divided into two parts: collaborative observation efficiency and emergency service capability. The synoptic observation efficiency refers to the overall working capability presented by the coordination among observation systems, which needs to take into consideration the inherent technical performance of heterogeneous sensors of dynamic scheduling and the degree of accomplishment of specific observation tasks by the synergy among platforms. The evaluation of emergency service capability refers to the dynamic performance of remote sensing
service systems in performing specific tasks, which is related to the application requirements of disaster emergency response.

### 4.1 Evaluation of effectiveness of coordinated space-air-ground remote sensing observations

A collaborative observation effectiveness assessment is a quantitative expression of the level of observation that remote sensing technology has in performing a particular task. This is closely related to the inherent properties of remote sensing technology and the type of task.

**4.1.1 Simulation calculations of coordination observation**

The collaborative observation system consists of multiple distributed remote sensor resource systems, and its technology enhances the observation capability in three aspects: data volume, accuracy and timeliness. The specific collaborative mode

is often determined according to the characteristics of remote sensing technology and the emergency needs of geological disasters. Taking the emergency observation of mudslide disaster as an example, the following two demands should be met:

(1) to quickly determine the scope of the disaster, (2) to quickly conduct disaster assessment and carry out rescue. Combining the advantage of wide observation range of satellite images and the ability of aerial remote sensing to deploy in real time, fly under clouds, be highly mobile and obtain data quickly, forming a typical synergistic way satellite-aerial: quickly obtain pre-disaster high-resolution remote sensing images to obtain geological information of the disaster area and initially determine the scope of the disaster to complete the pre-disaster research and judgment; combine post-disaster high-

resolution remote sensing images and aerial survey data for remote sensing interpretation to determine the disaster assessment base map, to provide decision support for rescue.

Based on the above analysis set, the following remote sensing synergistic approach formed through planning services after a mudslide disaster occurred in a certain place: (A) GF-2 satellite and the KC2600 UAV with Sony NEX-7 camera, (B) Pleiades satellite and the EWZ-D6 UAV with Nikon D800 camera and (C) IKONOS and DMC aerial cameras. Their

synergy effectiveness was calculated as follows:

(1) Identify evaluation indicators. Through the synergy of high-resolution satellites and aerial remote sensing, the emergency response time can be effectively shortened, the timeliness of data acquisition at disaster sites can be improved and effectiveness indicators can be established, as shown in Table 3.

**Table 3. Effectiveness indicators**

| Indicators | Implication |
|---|---|
| Spatial resolution of satellites | The higher the resolution, the more accurate the disaster information obtained |
| Flight preparation time | Time required for preflight arrangements of aircraft |
| Flight operating hours | Total hours of in-flight observation |
| Flight coverage | The larger the area covered, the more data exists. |
| Data processing | Data preprocessing capabilities |

(2) Table 3 contains two types of indicators: quantitative and qualitative. Quantitative indicators such as resolution can be obtained directly from the technical parameters of the satellite, while flight-related time and area coverage are obtained according to the technology of different products combined with operational experience. Data processing is a qualitative indicator for which we use 1 to represent having preprocessing capability and 2 for the opposite situation. The data of the sensor-type indicators involved in the collaborative observation scheme are extracted. A decision matrix $A$ is created,

trended and normalized to obtain matrix $B$.

$$A=\begin{bmatrix} 1 & 90 & 1.8 & 0.82 & 2 \\ 0.5 & 45 & 0.65 & 0.73 & 2 \\ 1 & 150 & 3.4 & 0.94 & 1 \end{bmatrix} \quad (13)$$

$$B=\begin{bmatrix} 0.41 & 0.43 & 0.46 & 0.57 & 0.41 \\ 0.82 & 0.86 & 0.17 & 0.51 & 0.41 \\ 0.41 & 0.26 & 0.87 & 0.65 & 0.82 \end{bmatrix} \quad (14)$$

(3) Use RSR to determine the weights. Table 4 lists the rank and RSR of the indicator values for each scenario, and Table 5 lists the final weights determined.

**Table 4. Indicator rank and RSR**

| Indicators | A | B | C | *RSR* |
|---|---|---|---|---|
| Spatial resolution of satellites | 1 | 4 | 2 | 0.47 |
| Flight preparation time | 3 | 5 | 1 | 0.60 |
| Flight operating hours | 4 | 1 | 5 | 0.67 |
| Flight coverage | 5 | 3 | 3 | 0.73 |
| Data processing | 2 | 2 | 4 | 0.53 |

**Table 5. Weighting of indicators**

| Indicators | SR | $W^{'}$ | $W$ |
|---|---|---|---|
| Spatial resolution of satellites | 0.16 | 0.25 | 0.20 |
| Flight preparation time | 0.20 | 0.16 | 0.16 |
| Flight operating hours | 0.22 | 0.18 | 0.20 |
| Flight coverage | 0.24 | 0.21 | 0.26 |
| Data processing | 0.18 | 0.20 | 0.18 |

(4) Calculate the weighted matrix $C$ to obtain PIS and NIS.

$$C=\begin{bmatrix} 0.08 & 0.07 & 0.09 & 0.15 & 0.07 \\ 0.16 & 0.14 & 0.03 & 0.13 & 0.07 \\ 0.08 & 0.04 & 0.18 & 0.17 & 0.13 \end{bmatrix} \quad (15)$$

$$C^{+} =\begin{bmatrix} 0.16 & 0.14 & 0.18 & 0.17 & 0.15 \end{bmatrix} \quad (16)$$

$$C^{-} =\begin{bmatrix} 0.08 & 0.04 & 0.03 & 0.13 & 0.07 \end{bmatrix} \quad (17)$$

(5) Calculation distance.

$$D^+ = \begin{bmatrix} 0.16 & 0.16 & 0.13 \end{bmatrix} \tag{18}$$

$$D^- = \begin{bmatrix} 0.07 & 0.13 & 0.16 \end{bmatrix} \tag{19}$$

(6) Calculation of the composite valuation.

$$S = \begin{bmatrix} 0.30 & 0.43 & 0.57 \end{bmatrix} \tag{20}$$

According to the evaluation results, the preferential order of the planning scheme is C, B and A; that is, the IKONOS and DMC aerial cameras have the strongest synergistic effect, the Pleiades satellite and EWZ-D6 UAV equipped with the Nikon D800 camera take second place and the GF-2 satellite and the KC2600 UAV equipped with the Sony NEX-7 camera have weak cooperation.

In this result, we analyze the main reason is: between the three, although the UAV flight preparation time of approach C is longer, but its operation time and coverage area are more advantageous, and these two indicators occupy a relatively large weight, respectively 0.2 and 0.26, while it has the ability of data processing, which leads to its final score is higher. Continuing to compare the two approaches A, B, on the basis of the same data processing capability, although the A approach has longer operation time and larger coverage area, B has excellent enough indicator values in the remaining two indicators (satellite resolution and flight preparation time) to make its final calculation result 0.43, which is higher than A's 0.3. This can show that the reasonableness of the setting of the weights has a very important position in influencing the accuracy of the results, and at the same time, it can make up for the deficiencies in other aspects when some indicators have outstanding advantages.

## 4.2 Evaluation of capacity of geohazard emergency response services

Emergency response to geohazards is a kind of disaster management that requires the coordination of multiple technologies for rescue and disaster relief. The top priority is to ensure personnel safety and save lives, and on this premise to avoid or reduce property losses to the greatest extent. In rescue work, there is a "golden 72 hours" during which the survival rate of the victims is extremely high. This is the critical rescue period after the occurrence of geological disasters. Remote sensing technology, as the main technical support for emergency response, should provide effective service for rescue in time and achieve fast investigation, fast characterization, fast decision-making and fast implementation of emergency work through cooperation. To evaluate the service capability of remote sensing collaborative systems in geohazard emergency responses, this section takes earthquake emergencies as an example, analyzes the demand to establish emergency response service chains and creates a Bayesian Network design evaluation model.

### 4.2.1. Earthquake emergency response service chain

Earthquakes are a sudden movement of the earth's surface caused by the release of slowly accumulating energy inside the earth that can cause substantial damage to life and property and further aggravate the impact of disasters and losses by triggering secondary disasters such as landslides, debris flows and barrier lakes. This paper refers to the process of remote sensing technology service in the Wenchuan earthquake emergency (PAN et al., 2010; Zhang et al., 2009), analyzes it from the aspects of spatial information demand for rapid response and information technology support for disaster relief and rescue and combines multiple remote sensing information services to form an earthquake remote sensing emergency service chain, as shown in Fig. 7. The need for emergency response to earthquakes is mainly reflected in the rapid acquisition of high-resolution remote sensing images, rapid processing of remote sensing data and extraction of hazard information. The "golden 72 hours" after an earthquake is a critical period for rescue, and high-resolution remote sensing images need to be quickly acquired and updated to analyze casualties, infrastructure damage, rescue and resettlement and other detailed information. The processing of remote sensing data in disaster relief needs to achieve real-time or near-real-time efficiency, including rapid image correction, alignment, stitching and uniform color. Disaster information is divided into three parts: building damage, lifeline damage and secondary disaster monitoring. Buildings reflect the main distribution of affected people. Roads are the lifeline of earthquake relief, and change analysis and feature extraction are the mainstay. These are combined with basic data and mathematical methods to analyze and calculate the scope of disaster impact and damage to buildings and roads and to make rapid assessments. Secondary disasters derived from earthquakes such as landslides, debris flows and barrier lakes are monitored dynamically by remote sensing technology and simulated to forecast their development and impact.

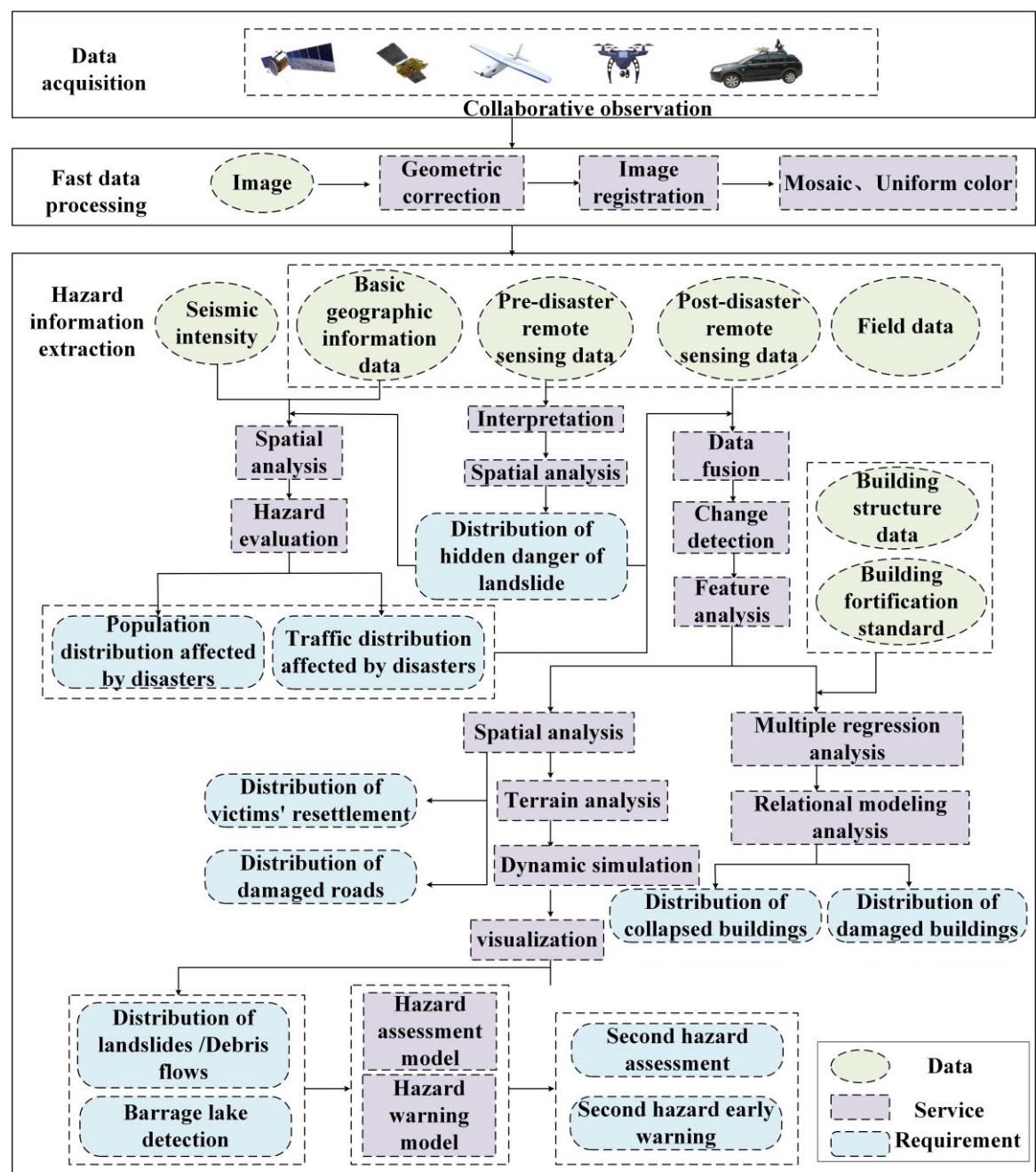

**Figure 7. Earthquake-remote sensing emergency response service chain**

### 4.2.2 Bayesian Network model for Emergency Services Evaluation

The Bayesian Network is a probabilistic graph model based on dependencies between variables, and its expected value is reliable when the causal chain is correct and has an appropriate probability distribution. The seismic emergency response service chain is a coherent link before and after, suitable for modeling by using a directed graph. The links are flexible, there

is uncertainty and the chain is suitable for handling with probability. Based on the theory established by Bayesian Network, the Bayesian Network model design for the evaluation of the earthquake disaster emergency response service capability was carried out using GeNIe Version 2.3 Academic software.

(1) Identify evaluation indicators. The system of indicators for evaluating the establishment of capacities according to the earthquake emergency response service chain is shown in Table 6.

**Table 6. Evaluation system for emergency response service capacity**

| First-level index | Second-level index | Third-level index |
| --- | --- | --- |
| Data acquisition | Planning | Response time |
| | | Reliability |
| | Observation | Technique |
| | | Range |
| | | Timeliness |
| Fast data processing | Correction | Accuracy |
| | Registration | Speed |
| | | Accuracy |
| | | Reliability |
| | Mosaic | Definition |
| | | Color equalization |
| | | Accuracy |
| Hazard information extraction | Function | Change detection |
| | | Spatial analysis |
| | | Terrain analysis |
| | Visualization | Expression |
| | | Timeliness |
| | | Reliability |
| | Forecast and Assessment | Content |
| | | Timeliness |
| | | Accuracy |

(2) Design the Bayesian Network structure. From the above evaluation system, the emergency response service capability is divided into three levels (data acquisition, fast data processing and disaster information extraction), and each level indicator is the parent node of the corresponding indicator of the previous level. This convergence relationship is represented by the directed edge from the parent node to the child node, i.e., from the lower-level indicators to the corresponding upper-level indicators that finally converge to the total indicators. Through the above analysis, the Bayesian Network topology of the

cooperative observation system earthquake emergency response service capability assessment model is constructed, as shown in Fig. 7.

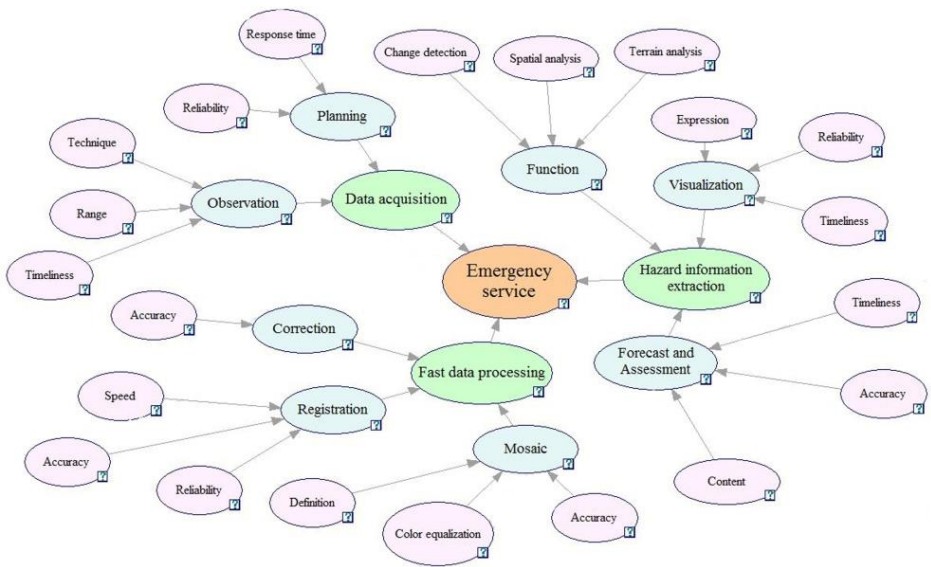

**Figure 8. Bayesian Network topology of service capacity evaluation mode**

(3) Construct the Bayesian Network model. Each node in the Bayesian Network model has a finite number of mutually exclusive states, where the root node is classified into three levels. The conditional probability of each node is determined according to expert experience to build the assessment model of the seismic emergency response service capability of the cooperative observation system, as shown in Fig. 8.

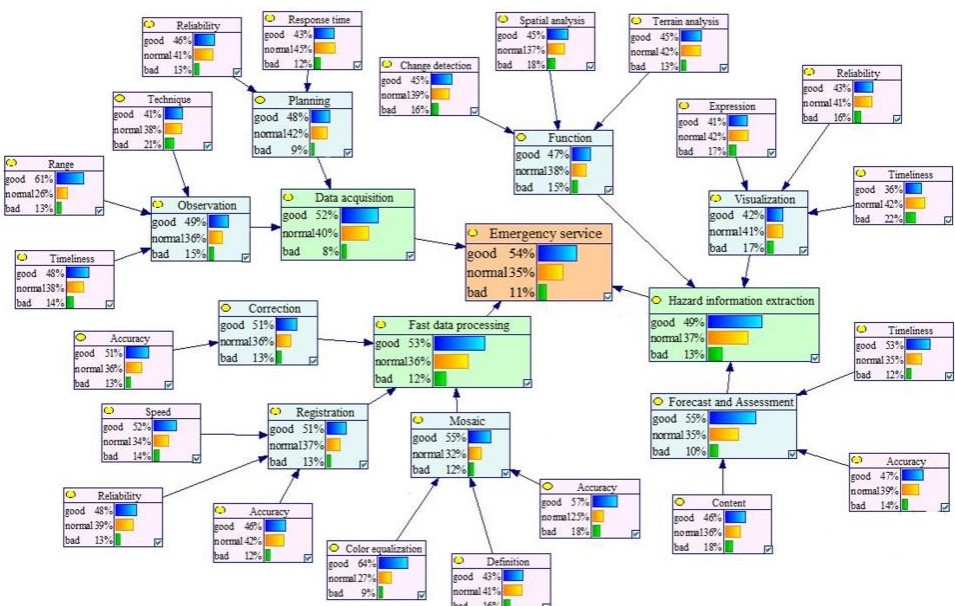

**Figure 9. Assessment model for capacity of collaborative observation system earthquake emergency response service**

(4) Capacity assessment. The capability of the cooperative observing system can be predicted by Bayesian inversion if the values of some nodes in the evaluation model are known. In setting the root node, the response time, observation range, correction accuracy, spatial analysis and forecast accuracy of the state are known (response time = good, observation range = normal, correction accuracy = good, spatial analysis = normal, forecast accuracy = normal). These are used as evidence variables to predict the capacity of the collaborative observation system, as shown in Fig. 9. Combined with Figure 8, after identifying the evidence variables, the probability of emergency response capacity being good increased from 54% to 60%, and the probabilities of the three first-level indexes were concentrated in good, good and normal: the probability of data acquisition ability being good increased from the previous 52% to 59%, the probability of fast data processing ability being good increased from 53% to 68%, however, the probability of the hazard information extraction ability being good decreased from 49% to 41%, and the probability of being normal increased from 37% to 45%. In this regard, it can be tentatively judged that the effectiveness of disaster emergency services can be further improved by improving the disaster information extraction link.

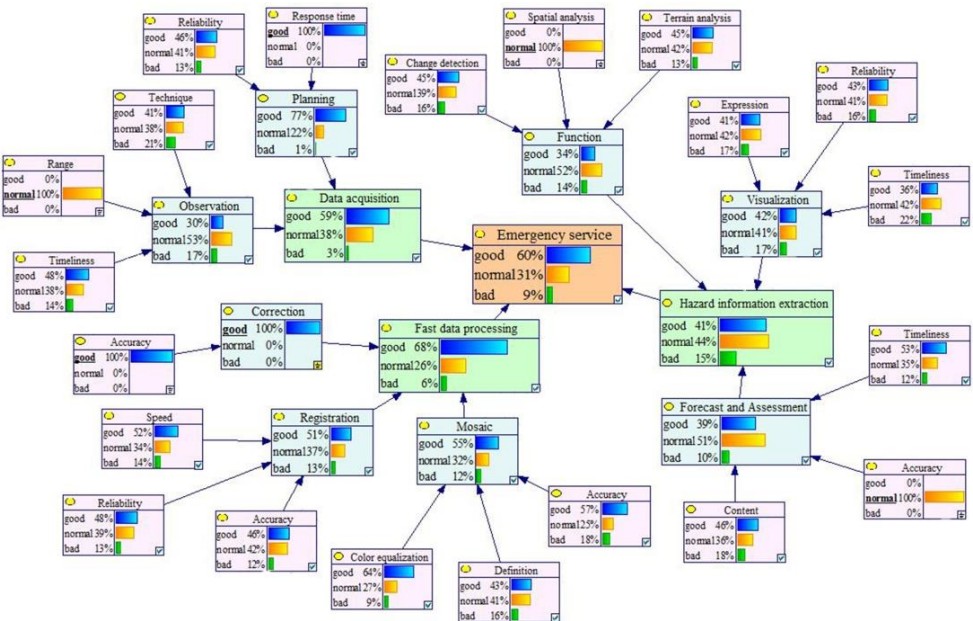

**Figure 10. Capacity projection of collaborative emergency services**

## 4.3. Discussion

This paper proposes a method for evaluating the synoptic observation effectiveness and emergency service capability of remote sensing using TOPSIS and Bayesian networks. The feasibility of the method is demonstrated by means of simulations in this chapter, but there are several situations that need to be addressed here.

(1) Determination of co-observation effectiveness indicators. The evaluation indexes are influenced by the performance of the sensors themselves and the cooperative mode. The index values related to the technical parameters of remote sensing can

be obtained directly from the database. These values include the spatial resolution of the satellite, revisit period, scan width, camera pixels and range of the laser scanner. Some index values need to be calculated in conjunction with the actual situation, including the flight height of the UAV and the flight coverage area.

(2) Determination of indicator weights in effectiveness evaluation. From the analysis of the results of the simulation experiments, it can be concluded that the index weights directly affect the results of the evaluation, and the setting of the weights reflects which aspect of the requirements the researcher cares more about, which will often be associated with the actual problem.

(3) Determination of rank division, probabilities and conditional probabilities of Bayesian Network nodes. In the above
calculation, the rank division, probability and conditional probability of each root node are the results of simulation statistics based on expert experience and can only be used to show that the evaluation network has computational feasibility and reference. In practical applications, these are difficult links to determine, and their accuracy directly affects the effectiveness of the Bayesian Network's work (Pourret O, 2008). The process of determination relies on a large amount of raw data as a reference for statistical analysis and requires a final value based on the actual application and combined with the opinions of
different experts. This needs to be further studied in our work.

(4) The uniqueness of Bayesian model design. The structure and node-level design of the Bayesian evaluation model are related to the demand for the application of the evaluation results, and the evaluation intention of the designer is indicated. In the above example, we considered the entire disaster response chain, which involved the effect of multiple aspects of data acquisition, processing and information extraction on disaster response. If one wants to examine the capability of only one
aspect of the emergency response, then the model can be designed separately. In addition, the complexity of actual disaster emergency response service links (and there are multiple design options for the same problem) need to be adjusted in the work according to the specific application and the needs of decision-makers.

(5) Dependence on the evaluation results. In the above study, due to the lack of real experimental scenario data, the study of disaster problems is only a generalized and simple calculation of geohazard simulation with reference to real events. The
results show that the evaluation can reflect the problems existing in the application of remote sensing technology. This is a reference for the planning of remote sensing cooperative observations in geohazard emergency work. However, an actual disaster emergency is a complex process, and the application of the method still needs to be revised in conjunction with a real situation.

## 5. Conclusions

This paper established a database of sensor technology and service indexes covering satellites, aviation and ground; realized the unified management of multiplatform and multitype heterogeneous sensor resources; and proposed a method to evaluate its application capability in geological disaster emergency response. This was accomplished by using TOPSIS and Bayesian networks in two aspects of collaborative observation effectiveness and emergency service capability, respectively. Thus, the

proposed method provides a decision basis for the establishment of air-space remote sensing collaborative services in

geological disaster emergency response.

Future work will include (1) further enriching the database content and developing Web service functions to realize the dynamic connection between data and evaluation calculation and (2) integrating more practical application scenarios and revising the evaluation calculation model.

**Author Contributions:** Liu, Y.H. and Zhang, J. designed the study. Liu, Y.H. performed the data collection, analysis, and

database establishment. Liu, Y.H. completed the design, calculations, and validation of the methods and models. Liu, Y.H. wrote the manuscript and led the revision with contributions from Zhang. J.  Zhang. J. managed the project schedule and budget.

**Acknowledgements:** This work was supported by the National Key Research and Development Program (No. 2018YFB0505402) and funded by the Natural Science Foundation of China (No.42171424).

**Conflicts of Interest:** The authors declare no conflict of interest.

**Appendix A**

**Table A1. Global satellite launch situation**

| Type of satellite | | Satellite | Country (area) |
|---|---|---|---|
| Land satellite | Landsat series | Landsat | USA |
| | | SPOT | France |
| | | CBERS | China-Brazil |
| | | ERS | ESA |
| | | ALMAZ | Russia |
| | | IRS | India |
| | | JERS | Japan |
| | High resolution satellite | IKONOS | USA |
| | | QuickBird | |
| | | WorldView-1/2/3/4 | |
| | | GeoEye1 | |
| | | Orbview3 | |
| | | SPOT-5/6/7 | France |
| | | Pleiades-1A/B | |
| | | RapidEye | Germany |

| | | ALOS | Japan |
|---|---|---|---|
| | | EROS-A/B | Israel |
| | | Resurs-DK1 | Russia |
| | | IRS-P7 | India |
| | | ZY-2 | China |
| | | GF-1/2/6/7 | |
| | | Formosat 2 | |
| | | Kompsat | Korea |
| | | THEOS | Thailand |
| | Hyper spectral satellite | EOS-AM1 | USA |
| | | EOS-PM1 | |
| | | EO-1 | |
| | | HJ-1A | China |
| | | GF-5 | |
| | SAR | ERS-1/2 | ESA |
| | | ENVISAT-1 | |
| | | TerraSAR-X | Germany |
| | | RADARSAT-1/2 | Canada |
| | | ALOS | Japan |
| | | COSMO-Sky Med | Italy |
| | | GF-3 | China |
| | | HJ-1C | |
| | Small satellite | SkySat | USA |
| | | LAPAN-Tubsat | Indonesia |
| | | BJ-1 | China |
| | | TH-1 | |
| | | SuperView-1 | |
| | | JL-1 | |
| | | OVS-1A/B | |
| | | TT-2 | |
| | | LJ-1 | |

| | | BNU-1 | |
|---|---|---|---|
| | | NOAA | USA |
| Meteorological satellite | | FY | China |
| | | GMS | Japan |
| | | SeaStar | USA |
| | | Jason | France |
| | | Sentinel-3 | ESA |
| Ocean satellite | | Okean | Russia |
| | | ADEOS | Japan |
| | | IRS-P4 | India |
| | | COMS-1 | Korea |
| | | HY | China |

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
