# Peer review of "Index establishment and capability evaluation of space-air-ground remote sensing cooperation in geohazard emergency response"

_Natural Hazards and Earth System Sciences, 2020_

## Referee Comment (RC1) · Anonymous Referee #1 · 30 Jan 2021

This study established an index database for various remote sensing data and analyzed the cooperative observation efficiency and emergency service capability. For the mitigation and rapid responses to geo-hazards, with various remote sensing tools, it is crucial to determine good combinations of the remote sensing tools and datasets. Therefore, this study demonstrates the architecture to establish the database and to determine the usage of good datasets, and it should be valuable for further applications and automation mechanisms. While it is a great work to establish such a database, there are some issues that the authors should consider. Especially, the

authors demonstrated that they can use the index database to evaluate the cooperative observation efficiency and emergency service capability via different models, but it will be better if the authors provide more analysis and validation to show that the evaluations they proposed are acceptable and appropriate.

In general, the writing of this paper is not that straight forward and could be more polished. In the main text, there are several abbreviations, but most of them are not well explained.

2.2. Indexes of technology and services: It seems that the indexes are the fundamental parameters of the database and analysis of this study. The authors could explain more why these indexes were chosen and justify if the indexes were appropriate and sufficient. What is the mechanism to increase (or decrease) the indexes?

Table 1: The technical indexes between each remote sensing types are not well separated.

3. Methodology: The authors mentioned some evaluation methods and used 2-3 of them in this study (the authors stated that TOPSIS and BN were used, but they further mentioned RSR was used as well, which is confusing). Here the authors could describe more rationale behind their choices (i.e. why they chose these methods over other methods? What are the advantages and disadvantages of these methods?)

Line 260: Terrestrial or ground mobile measurements provide in-situ observations that can be coupled with other type of remote sensing data. On the other hand, these measurements can also serve as ground truth data for validating other remote sensing data rather than equally play a role in the remote sensing synergies. I am wondering how this function of the terrestrial measurements is used and evaluated in the remote sensing coordination system?

4.1.2: The authors demonstrated an example of simulation calculations for determining better synergistic pair. Is there any ways to examine if the determination is reasonable?

Table 6: Similar to Table 1, the authors should put lines between different indexes in the tables.

4.2.2: Similar to 4.1.2, is it possible for the authors to qualitatively or quantitatively assess if their methods are reliable and appropriate? For example, the BN model shows the emergency response capacity increase to 60% in their example, but is there any other ways to validate this model result?

4.3: The title of this section is analysis, but I do not see much analysis here. Instead, the authors simply summarized their methods and results.

---

## Short Comment (SC1) · 15 Apr 2021

Thank you for your valuable comments, which have led to an improvement in the quality of our manuscript. Below you find a point-by-point reply to all specific questions and suggestions. Q 1: 2.2. Indexes of technology and services: It seems that the indexes are the fundamentalparameters of the database and analysis of this study. The authors could explainmore why these indexes were chosen and justify if the indexes were appropriate andsufficient. What is the mechanism to increase (or decrease) the
indexes? A 1: The main principle of the indicators selected in this paper is to be able to reflect the capabilities of different types of sensors, for which this study collects and summarizes the technical parameters of current types of sensors, and refers to the selection of indicators in some satellite online data repositories and the experience of relevant professionals in using them, on the basis of which universal sensor technology and service indicators are established. We will make changes in the manuscript as suggested. Q 2: Table 1: The technical indexes between each remote sensing types are not well separated. A2: In this regard, we will revise the table in the corresponding section of the manuscript to make its presentation clear. Q 3: 3. Methodology: The authors mentioned some evaluation methods and used 2-3 ofthem in this study (the authors stated that TOPSIS and BN were used, but they furthermentioned RSR was used as well, which is confusing). Here the authors could describemore rationale behind their choices (i.e. why they chose these methods over othermethods? What are the advantages and disadvantages of these methods?) A 3: In this paper, TOPSIS and BN are used as evaluation methods, but there is a weight determination in TOPSIS evaluation, and among the various ways of weight determination, RSR is used in this paper. this part of the paper will be revised in accordance with the recommendations, and the choice of methods will be explained. Q 4: 260: Terrestrial or ground mobile measurements provide in-situ observations thatcan be coupled with other type of remote sensing data. On the other hand, thesemeasurements can also serve as ground truth data for validating other remote sensingdata rather than equally play a role in the remote sensing synergies. I am wonderinghow this function of the terrestrial measurements is used and evaluated in the remotesensing coordination system? A 4: In fact your idea is very valuable and meaningful, and it is indeed an aspect that needs to be considered in a collaborative evaluation system, but in this paper we are mainly discussing the evaluation of capabilities in a disaster emergency environment, in which we are considering more the ability of ground measurement techniques to acquire data than their ability to validate data. Q 5: 4.1.2: The authors demonstrated an example of simulation calculations for determiningbetter synergistic pair. Is there any ways to

examine if the determination is reasonable? A 5: In fact we do lack an actual disaster application scenario to validate our experimental scheme for some reasons, so here we take simulations to illustrate the process of using the evaluation method proposed in this paper, and the results can be verified by expert experience, but this is a subjective method, and objectively there is still a need for real application situations to judge the results of the method, which is the shortage in our research and the direction to be strengthened afterwards. Q 6: Table 6: Similar to Table 1, the authors should put lines between different indexes inthe tables. A 6: As with question 2 we would follow the suggestions and revise the table in the appropriate section of the manuscript to make it clear. Q 7: 4.2.2: Similar to 4.1.2, is it possible for the authors to qualitatively or quantitativelyassess if their methods are reliable and appropriate? For example, the BN modelshows the emergency response capacity increase to 60% in their example, but is thereany other ways to validate this model result? A 7: There are numerous nodes and parameters involved in the BN model, and the determination of the ranking, probability and conditional probability of each node in the example of this paper is the result of simulation statistics and can only be used to show that the evaluation network is computationally feasible and informative. In practical applications, the determination of these data is a very important aspect, and its accuracy directly affects the working effect of Bayesian networks. The process of determination relies on a large amount of raw data as a reference for statistical analysis, and also requires a final value based on the actual application, combined with the opinions of different experts. Actual data in this area are still being accumulated, and this is the part that we hope to improve in subsequent studies. Q 8: 4.3: The title of this section is analysis, but I do not see much analysis here. Instead, the authors simply summarized their methods and results. A 8: We will revise and improve this part of the paper.

---

## Referee Comment (RC2) · Christian Bignami (Referee) · 16 May 2021

The work presents an analysis to provide a system to figure out the effectiveness and the capability of a remote sensing collaborative environment for emergency response to geo-hazards. I guess, but I'm not sure, this is an valuation of a so-called Decision Support System (DSS) but there is no mention about DSS.

First of all, the English must be deeply revised. There are many sentences are not clear, that hamper the understating of the work. Moreover, there are too repetition in the manuscript (e.g. *remote sensing cooperative*, or *space-air-ground remote sensing sensor*). The language revision is extremely important, otherwise, the massage carried on is lost. I found many difficulties on reading the manuscript. An example, first sentence really misleading: "*Geo-hazard emergency response is a disaster prevention and reduction action that multi-factorial, time-critical, task-intensive and socially significant*" ...emergency response cannot be prevention and reduction action.

A second important point is that the paper misses, since the beginning, indication on who is doing what for obtaining which results. This should be clear, immediately. The abstract does not allow to understand clearly whish are the objectives of the work, and does not present any figure of results

Last general and important comment: the manuscript has to be re-organised, is is not well presented. In many parts, the content of the section does not report appropriate information (see further comments), a reader can have difficulties on understanding the logic of the work.

Some other detailed issues, here in the following:

1. Instruction → I guess it is Introduction

Section 2, DATA.
Subsection 2.1: This part is not about the data used but a general discussion on the classification of sensors and some info on GIS. It presents an overview of the type of sensors, that is useful but not so much important to fill one page of the manuscript. I would expect to find here a focus on the way you set up the table A1 (that is not exhaustive and maybe can be neglected), giving the criteria adopted to build the table, and some number about the final database. Indeed, this initial part is linked with subsection 2.2 and 2.3
Abut Table A1: why do you list and consider satellites/missions that are no more working? E.g. ERS-1 and ERS-2, ALMAZ, JERS, IRS-P4….and others. It has no sense.
The Subsection 2.1 presents also a description of a GIS for emergency management. I think that this unit should be related to the main components of the system (or service) you want to analyse.
Fig.1 very small.
About the GIS emergency service: I think it is important to improve figure 2, trying to give information about the connections among all the modules/blocks in the scheme.

Section 2.2 and 2.3 I think they could merge because the indexes presented in table 1, are then used to build the database.
Figure 3, to small, and the content is not appreciable in the pdf provided for the review
Maybe split it into 2 figs, one for SAT_RS and one for SE_RS would help.
What is UML? No info about the acronym.

Section 3. Methodology
*The commonly used evaluation methods are*…for evaluating what? a Decision Support System? The performance of ??  what are we talking about?
After reading the simulation results I understand what you want to evaluate. You should say what you are evaluating here, considering you are giving some references of methods.
Please express all the acronym: AHP?? TOPSIS? Some are declared (RSR, BN) some other no. This also happens in subsections titles (RSR and Bayesian Network)

Section 3.1: how you determine the weighting vector W? I guess they are calculated using RSR, but there is no explanation on the relationship between RSR and TOPIS. It is almost clear only after reading sect. 3.2 and then at the simulation results section.
Section 3.2: in equation 9, are the elements Rij the same in matrix A (or B) of the section 3.1? not clear.
Why change meanings of m (objects) and n (indicators) indexes with respect matrix A (or B)?

Equation 10 and 11 are equal. What is SR term? And W' with respect W? Please explain.

Sect. 4 Results and discussion

The two subsection 4.1 and 4.1.1 are not related to presentation of results, but they seem a sort of introduction. The content repeats what already written before.

Section 4.1.2: can you give the criteria you adopt for selecting the remote sensing synergies (A), (B) and (C) for mudslide? I understand it is an example, but you have to justify this choice. Otherwise you can apply it also for earthquakes or other events.

How you set values in matrix A?

In table 4, in the first column two indicators have the same rank 1. One of them should be 2.

Section 4.2.1: figure 6 small, not well visible in the pdf provided for the review

Section 4.2.2: table 6. I would suggest to put horizontal lines for separating the related indexes. For example, Data Acquisition should be with Planning and Observation. These two are related to Response Time & Raliblity, and Technique+Range+Timeliness, repsctively. This will also help on reading figure 7.

About BN model? Hwa do you set the values for the third level (root) nodes?

Part number (4) (on page 18) seems an example of part number (3) (page 17), indeed figure 9 is the same of figure 8 excepting few values (maybe only the one of Forecast Accuracy. Hence, why duplicate it?

Finally, Section 4.3 Analysis: this is not an analysis section, rather a summary of the work with few comments at the end, in points (1), (2) and (3). This part must be improved and expanded, to figure out some issues and considerations about the results, the limits, the applicability, the selected environment for the example etc. etc.

Considering the comments provided above, that work cannot be published. I would propose the rejection of the manuscript, and encourage a resubmission.

---

## Author Comment (AC2) · 14 Jun 2021

Thank you for your valuable comments, which have led to an improvement in the quality of our manuscript. Below you find a point-by-point reply to all specific questions and suggestions.

**2.2. Indexes of technology and services: It seems that the indexes are the fundamentalparameters of the database and analysis of this study. The authors could explainmore why these indexes were chosen and justify if the indexes were appropriate andsufficient. What is the mechanism to increase (or decrease) the indexes?**

Re 1: The main principle of the indicators selected in this paper is to be able to reflect the capabilities of different types of sensors, for which this study collects and summarizes the technical parameters of current types of sensors, and refers to the selection of indicators in some satellite online data repositories and the experience of relevant professionals in using them, on the basis of which universal sensor technology and service indicators are established. We will make changes in the manuscript as suggested.

**Table 1: The technical indexes between each remote sensing types are not well separated.**

Re 2: In this regard, we will revise the table in the corresponding section of the manuscript to make its presentation clear.

**3. Methodology: The authors mentioned some evaluation methods and used 2-3 ofthem in this study (the authors stated that TOPSIS and BN were used, but they furthermentioned RSR was used as well, which is confusing). Here the authors could describemore rationale behind their choices (i.e. why they chose these methods over othermethods? What are the advantages and disadvantages of these methods?)**

Re 3: In this paper, TOPSIS and BN are used as evaluation methods, but there is a weight determination in TOPSIS evaluation, and among the various ways of weight determination, RSR is used in this paper. this part of the paper will be revised in accordance with the recommendations, and the choice of methods will be explained.

**Line 260: Terrestrial or ground mobile measurements provide in-situ observations thatcan be coupled with other type of remote sensing data. On the other hand, thesemeasurements can also serve as ground truth data for validating other remote sensingdata rather than equally play a role in the remote sensing synergies. I am wonderinghow this function of the terrestrial measurements is used and evaluated in the remotesensing coordination system?**

Re 4: In fact your idea is very valuable and meaningful, and it is indeed an aspect that needs to be considered in a collaborative evaluation system, but in this paper we are mainly discussing the evaluation of capabilities in a disaster emergency environment, in which we are considering more the ability of ground measurement techniques to acquire data than their ability to validate data.

**4.1.2: The authors demonstrated an example of simulation calculations for determiningbetter synergistic pair. Is there any ways to examine if the determination is reasonable?**

Re 5: In fact we do lack an actual disaster application scenario to validate our experimental scheme for some reasons, so here we take simulations to illustrate the process of using the evaluation method proposed in this paper, and the results can be verified by expert experience, but this is a subjective method, and objectively there is still a need for real application situations to judge the results of the method, which is the shortage in our research and the direction to be strengthened afterwards.

**Table 6: Similar to Table 1, the authors should put lines between different indexes inthe tables.**

Re 6: As with question 2 we would follow the suggestions and revise the table in the appropriate section of the manuscript to make it clear.

**4.2.2: Similar to 4.1.2, is it possible for the authors to qualitatively or quantitativelyassess if their methods are reliable and appropriate? For example, the BN modelshows the emergency response capacity increase to 60% in their example, but is thereany other ways to validate this model result?**

Re 7: There are numerous nodes and parameters involved in the BN model, and the determination of the ranking, probability and conditional probability of each node in the example of this paper is the result of simulation statistics and can only be used to show that the evaluation network is computationally feasible and informative. In practical applications, the determination of these data is a very important aspect, and its accuracy directly affects the working effect of Bayesian networks. The process of determination relies on a large amount of raw data as a reference for statistical analysis, and also requires a final value based on the actual application, combined with the opinions of different experts. Actual data in this area are still being accumulated, and this is the part that we hope to improve in subsequent studies.

**4.3: The title of this section is analysis, but I do not see much analysis here. Instead, the authors simply summarized their methods and results.**

Re 8: We will revise and improve this part of the paper.

---

## Author Response (AR1)

The authors would like to extend their sincere thanks to the referee for their time and considered thoughts on the submission. All comments and corrections have been thoroughly considered, with our respective action and/or response to these outlined below.

**Anonymous Referee #1:**

**Line 260: Terrestrial or ground mobile measurements provide in-situ observations that can be coupled with other type of remote sensing data. On the other hand, these measurements can also serve as ground truth data for validating other remote sensing data rather than equally play a role in the remote sensing synergies. I am wondering how this function of the terrestrial measurements is used and evaluated in the remote sensing coordination system?**

In fact your idea is very valuable and meaningful, and it is indeed an aspect that needs to be considered in a collaborative evaluation system, but in this paper we are mainly discussing the evaluation of capabilities in a disaster emergency environment, in which we are considering more the ability of ground measurement techniques to acquire data than their ability to validate data.

**4.1.2: The authors demonstrated an example of simulation calculations for determining better synergistic pair. Is there any ways to examine if the determination is reasonable?**

In fact we do lack an actual disaster application scenario to validate our experimental scheme for some reasons, so here we take simulations to illustrate the process of using the evaluation method proposed in this paper, and the results can be verified by expert experience, but this is a subjective method, and objectively there is still a need for real application situations to judge the results of the method, which is the shortage in our research and the direction to be strengthened afterwards.

**4.2.2: Similar to 4.1.2, is it possible for the authors to qualitatively or quantitatively assess if their methods are reliable and appropriate? For example, the BN model shows the emergency response capacity increase to 60% in their example, but is there any other ways to validate this model result?**

There are numerous nodes and parameters involved in the BN model, and the determination of the ranking, probability and conditional probability of each node in the example of this paper is the result of simulation statistics and can only be used to show that the evaluation network is computationally feasible and informative. In practical applications, the determination of these data is a very important aspect, and its accuracy directly affects the working effect of Bayesian networks. The process of determination relies on a large amount of raw data as a reference for statistical analysis, and also requires a final value based on the actual application, combined with the opinions of different experts. Actual data in this area are still being accumulated, and this is the part that we hope to improve in subsequent studies.

**Referee #2: Bignami, Christian**

**First of all, the English must be deeply revised. There are many sentences are not clear, that hamper the understating of the work. Moreover, there are too repetition in the manuscript (e.g. remote sensing cooperative, or space-air-ground remote sensing sensor). The language revision is extremely important, otherwise, the massage carried on is lost. I found many difficulties on reading the manuscript. An example, first sentence really misleading: "Geohazard emergency response is a disaster prevention and reduction action that multi-factorial, time-critical, task-intensive and socially significant" ...emergency response cannot be prevention and reduction action.**

**A second important point is that the paper misses, since the beginning, indication on who is doing what for obtaining which results. This should be clear, immediately. The abstract does not allow to understand clearly whish are the objectives of the work, and does not present any figure of results**

**Last general and important comment: the manuscript has to be re-organised, is is not well presented. In many parts, the content of the section does not report appropriate information (see further comments), a reader can have difficulties on understanding the logic of the work.**

The main purpose of this manuscript is to establish the capability evaluation system of space-air-ground remote sensing cooperative technology in terms of observation effectiveness and geohazard emergency response, so as to grasp its technical

operation and mission accomplishment, and provide a basis for decision making for space-air-ground remote sensing cooperative work. For the problems in the language expression and structure of the manuscript, we will make adjustments and modifications, and reorganize to highlight the focus of our doing work, so that it becomes more concise and logical.

**Section 2, DATA.**
**Subsection 2.1: This part is not about the data used but a general discussion on the classification of sensors and some info on GIS. It presents an overview of the type of sensors, that is useful but not so much important to fill one page of the manuscript. I would expect to find here a focus on the way you set up the table A1 (that is not exhaustive and maybe can be neglected), giving the criteria adopted to build the table, and some number about the final database. Indeed, this initial part is linked with subsection 2.2 and 2.3**
**Abut Table A1: why do you list and consider satellites/missions that are no more working? E.g. ERS-1 and ERS-2, ALMAZ, JERS, IRS-P4….and others. It has no sense.**
**The Subsection 2.1 presents also a description of a GIS for emergency management. I think that this unit should be related to the main components of the system (or service) you want to analyse.**
**Fig.1 very small.**
**About the GIS emergency service: I think it is important to improve figure 2, trying to give information about the connections among all the modules/blocks in the scheme.**
In this section 2.1, we mainly want to summarize the current development of space-air-ground remote sensing technology, and later analyze it to build a database. Table A1 is a brief list of remote sensing satellites launched by countries/regions according to their categories. Some satellites that are no longer working are listed because we think their technical parameters still have some reference, and we understand that the historical data of some of these satellites are still being used in some disaster emergencies. We will further revise the contents related to GIS in disaster emergencies, including Figures 1 and 2, to enhance the information expression.

**Section 2.2 and 2.3 I think they could merge because the indexes presented in table 1, are then used to build the database.**
**Figure 3, to small, and the content is not appreciable in the pdf provided for the review**
**Maybe split it into 2 figs, one for SAT_RS and one for SE_RS would help.**
**What is UML? No info about the acronym.**
For sections 2.2 and 2.3 we will merge the content, including Figure 3, which will enhance its information representation. UML-Unified Modeling Language, which is our unification of metrics for the purpose of creating a database, we will modify the corresponding section.

**Section 3. Methodology**
**The commonly used evaluation methods are…for evaluating what? a Decision Support System? The performance of ?? what are we talking about?**
**After reading the simulation results I understand what you want to evaluate. You should say what you are evaluating here, considering you are giving some references of methods.**
**Please express all the acronym: AHP?? TOPSIS? Some are declared (RSR, BN) some other no. This also happens in subsections titles (RSR and Bayesian Network)**
**Section 3.1: how you determine the weighting vector W? I guess they are calculated using RSR, but there is no explanation on the relationship between RSR and TOPIS. It is almost clear only after reading sect. 3.2 and then at the simulation results section.**
**Section 3.2: in equation 9, are the elements Rij the same in matrix A (or B) of the section 3.1? not clear.**
**Why change meanings of m (objects) and n (indicators) indexes with respect matrix A (or B)?**
**Equation 10 and 11 are equal. What is SR term? And W' with respect W? Please explain.**
We mainly want to evaluate the observation effectiveness of remote sensing technology and disaster emergency service capability respectively, and need to use some methods of system capability evaluation, the details of which we will strengthen in this section.
AHP- The Analytic Hierarchy Process, TOPSISI- Technique for Order Preference by Similarity to an Ideal Solution, the

expression of the abbreviation is an oversight in our manuscript, and we will further check and improve it.

There are various methods for determining the weights in the TOPSIS method, and in this paper we use the RSR (Rank-sum ratio) method, which we will explain in the corresponding section of the expression.( in section 3.1 of the revised manuscript, lines 228 through 237)

Equation 9 (in the revised manuscript into Equation 3) in Rij refers to the rank corresponding to the index value bij of the jth evaluation index in the ith evaluation object, different from the matrix in 3.1, for this part of the matrix A, B and the specific meaning of the index m, n we will strengthen the description to eliminate the expression of misunderstanding.

In Equations 10 and 11, SR (Score Ratio) is calculated from the RSR value of each evaluation indicator, referring to the proportional relationship between the levels of indicators, W' is the empirical weight, in order to somewhat eliminate the subjectivity of the evaluation, the final weight W is calculated using SR and W'.

**Sect. 4 Results and discussion**

**The two subsection 4.1 and 4.1.1 are not related to presentation of results, but they seem a sort of introduction. The content repeats what already written before.**

**Section 4.1.2: can you give the criteria you adopt for selecting the remote sensing synergies (A), (B) and (C) for mudslide? I understand it is an example, but you have to justify this choice. Otherwise you can apply it also for earthquakes or other events.**

**How you set values in matrix A?**

**In table 4, in the first column two indicators have the same rank 1. One of them should be 2.**

**Section 4.2.1: figure 6 small, not well visible in the pdf provided for the review**

**Section 4.2.2: table 6. I would suggest to put horizontal lines for separating the related indexes. For example, Data Acquisition should be with Planning and Observation. These two are related to Response Time & Raliblity, and Technique+Range+Timeliness, repsctively. This will also help on reading figure 7.**

**About BN model? Hwa do you set the values for the third level (root) nodes?**

**Part number (4) (on page 18) seems an example of part number (3) (page 17), indeed figure 9 is the same of figure 8 excepting few values (maybe only the one of Forecast Accuracy. Hence, why duplicate it?**

**Finally, Section 4.3 Analysis: this is not an analysis section, rather a summary of the work with few comments at the end, in points (1), (2) and (3). This part must be improved and expanded, to figure out some issues and considerations about the results, the limits, the applicability, the selected environment for the example etc. etc.**

The contents of 4.1 and 4.1.1 will be streamlined to reduce repetitive expressions.( Integrate the original 4.1.1 and 4.1.2 in the revised manuscript)

In 4.1.2 (4.1.1 in the revised manuscript), the selection of performance evaluation indicators is mainly based on the characteristics of synergistic technologies and observation needs, and we will explain the details of this part in the corresponding part of the manuscript.

The data in matrix A are of two types: qualitative and quantitative, for which qualitative data are obtained directly based on relevant information and experience, while quantitative data are defined and replaced by numbers, as we will explain further in the text in the manuscript.

We apologize for the error in the data in Table 4, it is a mistake in our work and we will revise it.

For Figure 6 and Table 6, we will revise them to make their presentation clear.

For the determination of the root node in the BN model, theoretically, it is necessary to determine its a priori data and then correct it by parameter learning, but due to the lack of data in this area, parameter learning is not possible, and we determine the data in this part mainly by expert experience.

Figure 8 shows the initial evaluation network model established, and Figure 9 shows the calculation results with certain nodes in the evaluation model taking values (setting them as evidence variables), which are illustrated by the change of data in the evaluation model to have a predictive assessment and guidance for the disaster emergency process.

Section 4.3 does have many shortcomings, thank you for your valuable comments, we will make further improvements.

[revised manuscript text omitted]

**Commented [4]:** Referee #2: Bignami, Christian: About the GIS emergency service: I think it is important to improve figure 2, trying to give information about the connections among all the modules/blocks in the scheme.

**Commented [5]:** Referee #2: Bignami, Christian: Section 2.2 and 2.3 I think they could merge ...

divided into two parts: SAT_RS, the sensor technical performance index database;and SE_RS, the emergency service evaluation index database.

285 The technical performance indicators of sensors in SAT_RS are their various capability characteristics under normal operation as reflected by technical parameters. The indicators are independent between different types of sensors. The parameters vary, and the technical indicators are also diverse. In the face of complex geohazard emergency response needs, how to select the appropriate sensors to accomplish the observation tasks requires the classification of existing sensors according to their capabilities and a synthesis of technical indicators. In this regard, this study collected and summarized the

290 technical parameters of various types of sensors, referred to the selection of indicators in satellite online data repositories (NSSDC 2020, CEOS 2020 and OSCAR 2020) and the experience of relevant professionals in using them, analyzed the information of various types of sensors and established a more complete sensor technology index system. This is shown in Table 1 below.

The indicators in the table are mainly considered in terms of the amount of information, timeliness, validity (accuracy) and

295 expressiveness of data acquisition. The indicators are selected for different types of sensor technical indicators. The amount of information is used to eliminate uncertainty, reflecting the intensity of the acquired information, and is related to the breadth and depth of the sensor's role with regard to the scan width, side-swing capability, measurement range, etc. Timeliness refers to the self-conscious dynamism of the sensor system and the degree of sensitivity and response to the task, and is related to the responsiveness and execution efficiency of the sensor. Factors include the revisit cycle of the satellite,

300 preparation time of the UAV and endurance. Validity expresses the accuracy of the acquired information with regard to resolution, quantization level and measurement accuracy, for example. Expressiveness describes the representational form of the information. Note that the same indicator has multiple effects on data acquisition, such as the spatial resolution of the satellite having an impact on both the amount and validity of information. Thus, it is necessary to set a comprehensive evaluation indicator of information acquisition capability in different dimensions.

305 **Table 1. Sensor technical indexes**

| Type | Technical indexes |
|------|------|
| Optical satellite | Spatial resolution |
| | Spectral resolution |
| | Radiation resolution |
| | Revisit time |
| | Swinging ability |
| | Swath width |
| SAR satellite | Wave band |
| | Polarization |
| | Spatial resolution |

Commented [6]: Anonymous Referee #1:2.2. Indexes of technology and services: It seems that the indexes are the fundamental parameters of the database and analysis of this study. The authors could explain more why these indexes were chosen and justify if the indexes were appropriate and sufficient. What is the mechanism to increase (or decrease) the indexes?

Commented [7]: Anonymous Referee #1:Table 1: The technical indexes between each remote sensing types are not well separated.

| | |
|---|---|
| | Revisit time |
| | Swath width |
| | Incidence angle |
| Photogrammetry | Resolution |
| | Data type |
| | Preparation time (UAV) |
| LiDAR | Point cloud density |
| | Measuring range |
| | Measurement accuracy |
| UAV | Endurance time |
| | Cruising speed |
| | Payload |
| Mobile measurement | Measuring range |
| | Data type |

The emergency service indexes in SE_RS refer to the capacity evaluation indexes of the emergency service system associated with the event. The space-air-ground remote sensing geohazard emergency service capacity evaluation index system established is shown in Table 2 below. The index is measured in the three aspects of data acquisition, processing and information service. The specific content of the indexes should be determined in conjunction with the responding emergency event.

**Table 2. Indexes for evaluation of emergency service capacity**

| Constitute | Criterion |
|---|---|
| Data acquisition | Technology, Data Volume, Timeliness, Responsiveness |
| Data processing | Methodology, Speed, Quality |
| Information services | Demand, Quality, Timeliness |

The database design process is divided into information analysis, structure design, storage settings and data storage. By analyzing the massive sensor information, we set the attribute fields from the carrying platform, set the technical characteristics of each type of sensor and operation status, store the corresponding data in the database and establish unified management and finally design 20 kinds of tables. This is shown in Fig. 3. SAT_RS records the basic satellite, aerial and terrestrial information through three tables: RS_Satellite, Sensor_Aerial and RS_Terrestrial. Then, SAT_RS establishes a technical characteristics index table for different types of sensors on different platforms according to the technical index system shown in Table 1. This includes SatelliteSensor_Optical, SatelliteSensor_SAR, UAS, ImageSpectrometer, DigitalCamera, AirbronrLiDAR, MinSAR, MMS and 11 other types of tables. The RS_Task table in SE_RS links tasks among sensors and records the observation tasks they perform, including observation equipment and observation time. The

evaluation indexes in RS_DataProcessing and RS_Service are set according to the guidelines of Table 2 and the specific geohazard remote sensing emergency service events.

[Figure]

**Figure 3. Unified Modeling Language (UML) of database**

Commented [8]: Referee #2: Bignami, Christian:a: Figure 3, to small. b: What is UML?

325    At present, the SAT_RS database records approximately 150 satellites and their corresponding sensor data from many countries and organizations including the United States, France, ESA, Russia, Japan, Korea, India and China; more than 100 commonly used aerial remote sensor product families; more than 50 UAV products; and dozens of ground mobile measurement systems. A partial display is shown in Fig. 4. The features of SAT-RS are as follows: (1) Wide data coverage, support for satellites, aviation platforms (including UAVs), terrestrial multiplatforms and multiple types of remote sensors.

330    (2) Indexing of sensor technical performance and support for evaluation calculations. (3) Data support for sensor ML descriptions.

[Figure]

**Figure 4. SAT_RS database (partial)**

**3 Methodology**

335   The methods commonly used to evaluate the system capabilities are the analytic hierarchy process (AHP) (Emrouznejad et al., 2017), fuzzy integrated assessment (Kahraman et al., 2015), technique for order preference by similarity to an ideal solution (TOPSIS) (Zhang, 2015), rank sum ratio (RSR) (Tian, 2002) and Bayesian network (BN) (Ejsing et al., 2008), all of which have their own characteristics. The evaluation studied in this paper is a complex and flexible multisystem and multi-influencing factor problem. In order to improve the scientific nature of the evaluation and make full use of the advantages of

340   various methods, TOPSIS and Bayesian-network-based evaluation methods are used for remote sensing collaborative observation and service capability, respectively, while RSR is used to determine the weights in TOPSIS calculation. The evaluation process is shown in Figure 5 below.

TOPSIS can eliminate the influence of different indicator magnitudes and make full use of the information of the original data. This is a common method for multiobjective decision analysis of limited solutions in systems engineering. Since this

345   method has no strict restrictions on the distribution, quantity and magnitudes of evaluation data, it is flexible in application and can be well adapted to the changes of indicators involving many types of sensors. Meanwhile, the use of RSR to determine the weights combines the Score Ratio (SR) and empirical weights. This overcomes to some extent the subjectivity of determining the weights and makes the evaluation results more reflective of objective facts.

A BN is a probabilistic graphical model based on the dependency relationships among variables, and the evaluation of

350   emergency response service capability using this method has the following advantages. First, the emergency response process can be divided into a number of coherent and causally related links such as data acquisition, processing, information extraction and forming an emergency service chain. The evaluation of this service capability with backward and forward

**Commented [9]:** Referee #2: Bignami, Christian:... for evaluating what?

**Commented [10]:** Referee #2: Bignami, Christian: Please express all the acronym: AHP?? TOPSIS?

correlation is suitable for modeling with directed graphs. Second, there are uncertainties in each link of emergency services that are suitable for probabilistic methods.

**Commented [11]:** Anonymous Referee #1: 3. Methodology: The authors mentioned some evaluation methods and used 2-3 of them in this study (the authors stated that TOPSIS and BN were used, but they further mentioned RSR was used as well, which is confusing). Here the authors could describe more rationale behind their choices (i.e. why they chose these methods over other methods? What are the advantages and disadvantages of these methods?)

[revised manuscript text omitted]

**Commented [15]:** Referee #2: Bignami, Christian: Section 4.1.2: can you give the criteria you adopt for selecting the remote sensing synergies (A), (B) and (C) for mudslide?

**Commented [16]:** Referee #2: Bignami, Christian: How you set values in matrix A?

(3) Use RSR to determine the weights. Table 4 lists the rank and RSR of the indicator values for each scenario, and Table 5 lists the final weights determined.

**Table 4. Indicator rank and RSR**

| Indicators | A | B | C | *RSR* |
|---|---|---|---|---|
| Spatial resolution of satellites | 1 | 4 | 2 | 0.47 |
| Flight preparation time | 3 | 5 | 1 | 0.60 |
| Flight operating hours | 4 | 1 | 5 | 0.67 |
| Flight coverage | 5 | 3 | 3 | 0.73 |
| Data processing | 2 | 2 | 4 | 0.53 |

**Commented [17]: Referee #2: Bignami, Christian:** In table 4, in the first column two indicators have the same rank 1. One of them should be 2.

**Table 5. Weighting of indicators**

| Indicators | SR | $W^{'}$ | $W$ |
|---|---|---|---|
| Spatial resolution of satellites | 0.16 | 0.25 | 0.20 |
| Flight preparation time | 0.20 | 0.16 | 0.16 |
| Flight operating hours | 0.22 | 0.18 | 0.20 |
| Flight coverage | 0.24 | 0.21 | 0.26 |
| Data processing | 0.18 | 0.20 | 0.18 |

(4) Calculate the weighted matrix $C$ to obtain PIS and NIS.

$$C=\begin{bmatrix} 0.08 & 0.07 & 0.09 & 0.15 & 0.07 \\ 0.16 & 0.14 & 0.03 & 0.13 & 0.07 \\ 0.08 & 0.04 & 0.18 & 0.17 & 0.13 \end{bmatrix} \tag{15}$$

$$C^{+}=\begin{bmatrix} 0.16 & 0.14 & 0.18 & 0.17 & 0.15 \end{bmatrix} \tag{16}$$

$$C^{-}=\begin{bmatrix} 0.08 & 0.04 & 0.03 & 0.13 & 0.07 \end{bmatrix} \tag{17}$$

(5) Calculation distance.

$$D^{+}=\begin{bmatrix} 0.16 & 0.16 & 0.13 \end{bmatrix} \tag{18}$$

$$D^{-}=\begin{bmatrix} 0.07 & 0.13 & 0.16 \end{bmatrix} \tag{19}$$

(6) Calculation of the composite valuation.

$$S=\begin{bmatrix} 0.30 & 0.43 & 0.57 \end{bmatrix} \tag{20}$$

According to the evaluation results, the preferential order of the planning scheme is C, B and A; that is, the IKONOS and DMC aerial cameras have the strongest synergistic effect, the Pleiades satellite and EWZ-D6 UAV equipped with the Nikon D800 camera take second place and the GF-2 satellite and the KC2600 UAV equipped with the Sony NEX-7 camera have weak cooperation.

455 **4.2 Evaluation of capacity of geohazard emergency response services**

Emergency response to geohazards is a kind of disaster management that requires the coordination of multiple technologies for rescue and disaster relief. The top priority is to ensure personnel safety and save lives, and on this premise to avoid or reduce property losses to the greatest extent. In rescue work, there is a "golden 72 hours" during which the survival rate of the victims is extremely high. This is the critical rescue period after the occurrence of geological disasters. Remote sensing

460 technology, as the main technical support for emergency response, should provide effective service for rescue in time and achieve fast investigation, fast characterization, fast decision-making and fast implementation of emergency work through cooperation. To evaluate the service capability of remote sensing collaborative systems in geohazard emergency responses, this section takes earthquake emergencies as an example, analyzes the demand to establish emergency response service chains and creates a BN design evaluation model.

465 **4.2.1. Earthquake emergency response service chain**

Earthquakes are a sudden movement of the earth's surface caused by the release of slowly accumulating energy inside the earth that can cause substantial damage to life and property and further aggravate the impact of disasters and losses by triggering secondary disasters such as landslides, debris flows and barrier lakes. This paper refers to the process of remote sensing technology service in the Wenchuan earthquake emergency (PAN et al., 2010; Zhang et al., 2009), analyzes it from the

470 aspects of spatial information demand for rapid response and information technology support for disaster relief and rescue and combines multiple remote sensing information services to form an earthquake remote sensing emergency service chain, as shown in Fig. 6. The need for emergency response to earthquakes is mainly reflected in the rapid acquisition of high-resolution remote sensing images, rapid processing of remote sensing data and extraction of hazard information. The "golden 72 hours" after an earthquake is a critical period for rescue, and high-resolution remote sensing images need to be quickly

475 acquired and updated to analyze casualties, infrastructure damage, rescue and resettlement and other detailed information. The processing of remote sensing data in disaster relief needs to achieve real-time or near-real-time efficiency, including rapid image correction, alignment, stitching and uniform color. Disaster information is divided into three parts: building damage, lifeline damage and secondary disaster monitoring. Buildings reflect the main distribution of affected people. Roads are the lifeline of earthquake relief, and change analysis and feature extraction are the mainstay. These are combined with

480 basic data and mathematical methods to analyze and calculate the scope of disaster impact and damage to buildings and roads and to make rapid assessments. Secondary disasters derived from earthquakes such as landslides, debris flows and

barrier lakes are monitored dynamically by remote sensing technology and simulated to forecast their development and impact.

[Figure]

**Figure 7. Earthquake-remote sensing emergency response service chain**

485

**Commented [18]: Referee #2: Bignami, Christian**:Figure 6 small.

**4.2.2 BN model for Emergency Services Evaluation**

BN is a probabilistic graph model based on dependencies between variables, and its expected value is reliable when the causal chain is correct and has an appropriate probability distribution. The seismic emergency response service chain is a coherent link before and after, suitable for modeling by using a directed graph. The links are flexible, there is uncertainty and the chain is suitable for handling with probability. Based on the theory established by BN, the BN model design for the evaluation of the earthquake disaster emergency response service capability was carried out using GeNIe Version 2.3 Academic software.

(1) Identify evaluation indicators. The system of indicators for evaluating the establishment of capacities according to the earthquake emergency response service chain is shown in Table 6.

**Table 6. Evaluation system for emergency response service capacity**

| First-level index | Second-level index | Third-level index |
|---|---|---|
| Data acquisition | Planning | Response time |
| | | Reliability |
| | Observation | Technique |
| | | Range |
| | | Timeliness |
| Fast data processing | Correction | Accuracy |
| | Registration | Speed |
| | | Accuracy |
| | | Reliability |
| | Mosaic | Definition |
| | | Color equalization |
| | | Accuracy |
| Hazard information extraction | Function | Change detection |
| | | Spatial analysis |
| | | Terrain analysis |
| | Visualization | Expression |
| | | Timeliness |
| | | Reliability |
| | Forecast and Assessment | Content |
| | | Timeliness |
| | | Accuracy |

Commented [19]: Anonymous Referee #1: Table 6: Similar to Table 1, the authors should put lines between different indexes in the tables. Referee #2: Bignami, Christian: Section 4.2.2: table 6. I would suggest to put horizontal lines for separating the related indexes.

(2) Design the BN structure. From the above evaluation system, the emergency response service capability is divided into three levels (data acquisition, fast data processing and disaster information extraction), and each level indicator is the parent node of the corresponding indicator of the previous level. This convergence relationship is represented by the directed edge from the parent node to the child node, i.e., from the lower-level indicators to the corresponding upper-level indicators that finally converge to the total indicators. Through the above analysis, the BN topology of the cooperative observation system earthquake emergency response service capability assessment model is constructed, as shown in Fig. 7.

[Figure]

**Figure 8.  BN topology of service capacity evaluation mode**

(3) Construct the BN model. Each node in the BN model has a finite number of mutually exclusive states, where the root node is classified into three levels. The conditional probability of each node is determined according to expert experience to build the assessment model of the seismic emergency response service capability of the cooperative observation system, as shown in Fig. 8.

Commented [20]: Referee #2: Bignami, Christian: About BN model? Hwa do you set the values for the third level (root) nodes?

[Figure]

**Figure 9. Assessment model for capacity of coordinated observation system earthquake emergency response service**

510 (4) Capacity assessment. The capability of the cooperative observing system can be predicted by Bayesian inversion if the values of some nodes in the evaluation model are known. In setting the root node, the response time, observation range, correction accuracy, spatial analysis and forecast accuracy of the state are known (response time = good, observation range = normal, correction accuracy = good, spatial analysis = normal, forecast accuracy = normal). These are used as evidence variables to predict the capacity of the collaborative observation system, as shown in Fig. 9. The emergency response 515 capacity of good probability increased to 60%, and the probability of hazard information extraction is concentrated in the normal. This can be initially judged by improving the disaster information extraction link to further improve the effectiveness of disaster emergency services.

[Figure]

**Figure 10. Capacity projection of collaborative emergency services**

**4.3. Discussion**

This paper proposes a method for evaluating the synoptic observation effectiveness and emergency service capability of remote sensing using TOPSIS and Bayesian networks. The feasibility of the method is demonstrated by means of simulations in this chapter, but there are several situations that need to be addressed here.

(1) Determination of co-observation effectiveness indicators. The evaluation indexes are influenced by the performance of the sensors themselves and the cooperative mode. The index values related to the technical parameters of remote sensing can be obtained directly from the database. These values include the spatial resolution of the satellite, revisit period, scan width, camera pixels and range of the laser scanner. Some index values need to be calculated in conjunction with the actual situation, including the flight height of the UAV and the flight coverage area.

(2) Determination of rank division, probabilities and conditional probabilities of BN nodes. In the above calculation, the rank division, probability and conditional probability of each root node are the results of simulation statistics based on expert experience and can only be used to show that the evaluation network has computational feasibility and reference. In practical applications, these are difficult links to determine, and their accuracy directly affects the effectiveness of the BN's work (Pourret O, 2008). The process of determination relies on a large amount of raw data as a reference for statistical analysis and requires a final value based on the actual application and combined with the opinions of different experts. This needs to be further studied in our work.

(3) The uniqueness of Bayesian model design. The structure and node-level design of the Bayesian evaluation model are related to the demand for the application of the evaluation results, and the evaluation intention of the designer is indicated. In

the above example, we considered the entire disaster response chain, which involved the effect of multiple aspects of data acquisition, processing and information extraction on disaster response. If one wants to examine the capability of only one aspect of the emergency response, then the model can be designed separately. In addition, the complexity of actual disaster emergency response service links (and there are multiple design options for the same problem) need to be adjusted in the work according to the specific application and the needs of decision-makers.

(4) Dependence on the evaluation results. In the above study, due to the lack of real experimental scenario data, the study of disaster problems is only a generalized and simple calculation of geohazard simulation with reference to real events. The results show that the evaluation can reflect the problems existing in the application of remote sensing technology. This is a reference for the planning of remote sensing cooperative observations in geohazard emergency work. However, an actual disaster emergency is a complex process, and the application of the method still needs to be revised in conjunction with a real situation.

> **Commented [21]:** Anonymous Referee #1:The title of this section is analysis, but I do not see much analysis here. Instead, the authors simply summarized their methods and results.
> **Referee #2: Bignami, Christian:** Section 4.3 Analysis: this is not an analysis section, rather a summary of the work with few comments at the end, in points (1), (2) and (3). This part must be improved and expanded, to figure out some issues and considerations about the results, the limits, the applicability, the selected environment for the example etc. etc.

[revised manuscript text omitted]

---

## Referee Report (RR1)

This study aims to evaluate the multi-sensor collaborative observation and service capability, which is important and valuable to the better selection and planning of sensors for geographical disaster observation. I have read the former comments and revisions, and I can see significant improvements in this paper. Although I think this paper is close to success, I have some concerns and questions:

In the Introduction, the materials of the collaboration of heterogeneous sensors are kind of old, and it seems that some latest works are missing. Just to name a few:

Hu, et al., 2019. An Observation Capability Information Association Model for Multisensor Observation Integration Management: A Flood Observation Use Case in the Yangtze River Basin. IEEE Sensors Journal, 19(3), 11510-11525.

Hu, et al., 2020. Observation Capability Representation for GeoTask-Oriented Multi-Sensor Planning Cognition. *International Journal of Geographical Information Science*, 32(2), 205-228.

Wang, et al., 2020. A Collaborative Planning Method of Space-Ground Sensor Network Coverage Optimization for Multiparameter Observation Tasks. *IEEE Sensors Journal*, 21(6), 8384-8399.

In Figure 1, I think it is better to number all lines but not just "observation-transmission-process-distribution", which could provide the reader a better understanding of the workflow.

Line 123. "WMP" should be "WMS"; line 125. "… visualization studies…", it seems that the "studies" used here is a misspelling.

Lines 152-153. "The amount of information is used to eliminate uncertainty …", what uncertainty exists, and what uncertainty you are trying to eliminate?

In section 3.1, you have introduced the detailed calculations of TOPSIS step by step. However, I think it could be clearer if you introduced it in conjunction with the collaborative observation capability assessment background. For example, what is the meaning of each element in the decision matrix, and what the relative closeness reflects?

Line 279. "The specific collaborative mode …", the collaborative mode refers to what is not explicitly given here.

Line 287. Although this experiment is a simulation, I would like to know whether there have any specific task/emergency observation requirements (e.g., task observation space and time). Besides, I think the ability to observe the task area is a fundamental requirement. Since the satellite flying around the earth all the time, whether a satellite can monitor a specific area requires additional calculation. Thus, I would also like to know how those satellites and UAVs are retrieved from your database.

In Table 3, the first indicator is "Spatial resolution of satellites", why not the UAV or the sensor? Besides, each collaborated satellite and UAV mounted the same sensor, what you would do if their mounted sensors were different?

Line 329. The mentioned Figure should be Figure 7.

Line 372. The result here is not clear. You claimed that the emergency response capability of good increased to 60%, but you didn't give the probability before the increase. Moreover, it seems that the third-level indexes should be sourced from satellites and UAVs, but those satellites or UAVs are not given.

Lastly, I think there lacks a criterion analysis and discussion of the experimental results. For example, although the scores of A, B, and C (section 4.1) are calculated, the reason why C is the best is not analyzed and discussed.

---

## Author Response (AR2)

The authors would like to extend their sincere thanks to the referee for their time and considered thoughts on the submission. All comments and corrections have been thoroughly considered with our respective action and/or response to these outlined below.

5 In the Introduction, the materials of the collaboration of heterogeneous sensors are kind of old, and it seems that some latest works are missing.

Thank you for your valuable comments, which were an oversight in our work, and this part of the literature has been updated in the new manuscript. The added references are as follows. (Line56-59, Line77-82)

 [1] Haghighi, M. H., Motagh, M.: Ground surface response to continuous compaction of aquifer system in Tehran, Iran:
 Results from a long-term multi-sensor InSAR analysis, Remote Sensing of Environment, 221:534-550. https://doi.org/10.1016/j.rse.2018.11.003, 2019.

[2] Hermle, D., Keuschnig, M., Hartmeyer, I., et al.: Timely prediction potential of landslide early warning systems with multispectral remote sensing: a conceptual approach tested in the Sattelkar, Austria, Nat. Hazards Earth Syst. Sci., 21, 2753–2772, https://doi.org/10.5194/nhess-21-2753-2021, 2021.

- [3] Lu, P., Qin, Y., Li, Z., et al.: Landslide mapping from multi-sensor data through improved change detection-based Markov random field. Remote Sensing of Environment, 231:111235. https://doi.org/10.1016/j.rse.2019.111235, 2019.
  [4] Hu, C.L., Tian, L., Li, J., et al.: An Observation Capability Information Association Model for Multisensor Observation Integration Management: A Flood Observation Use Case in the Yangtze River Basin, IEEE sensors journal, 19(23):11510-11525. https://doi.org/10.1109/JSEN.2019.2933655, 2019.
- 20 [5] Hu, C.L., Li, J., Xiao, C., et al.: SOCO-Field: observation capability representation for GeoTask-oriented multi-sensor planning cognition. International Journal of Geographical Information Science, 1-24. https://doi.org/10.1080/13658816.2019.1655755, 2020.

 [6] Wang, K., et al.: A Collaborative Planning Method of Space-Ground Sensor Network Coverage Optimization for Multiparameter Observation Tasks. IEEE Sensors Journal, 21(6), 8384-8399. https://doi.org/10.1109/JSEN.2020.3048035.2020.

In Figure 1, I think it is better to number all lines but not just "observation-transmission-process-distribution", which could provide the reader a better understanding of the workflow.

Figure 1 was modified according to the suggestions to make the expression clearer. (Line118-121, Line123)

30

25

Line 123. "WMP" should be "WMS"; line 125. "... visualization studies...", it seems that the "studies" used here is a misspelling.

Corrections were made to the corresponding parts of the manuscript. (Line129, Line131)

Lines 152-153. "The amount of information is used to eliminate uncertainty ...", what uncertainty exists, and what uncertainty you are trying to eliminate?

The amount of information is used to remove the uncertainty in the observed data, here mainly the uncertainty present in the representation of spatio-temporal information, to determine the temporal extent, the spatial area, the degree of spatial detail such as geometry and properties, etc. The manuscript has been refined in the corresponding section. (Line158-161)

40 In section 3.1, you have introduced the detailed calculations of TOPSIS step by step. However, I think it could be clearer if you introduced it in conjunction with the collaborative observation capability assessment background. For example, what is the meaning of each element in the decision matrix, and what the relative closeness reflects? Thank you very much for your comments, we have revised the methodological introduction of TOPSIS in section 3.1.

45 Line 279. "The specific collaborative mode ...", the collaborative mode refers to what is not explicitly given here.

The specific collaborative mode is satellite-aerial: quickly obtain pre-disaster high-resolution remote sensing images to obtain geological information of the disaster area and initially determine the scope of the disaster to complete the pre-disaster research and judgment; combine post-disaster high-resolution remote sensing images and aerial survey data for remote sensing interpretation to determine the disaster assessment base map, to provide decision support for rescue. The manuscript has been revised and improved in the corresponding parts. (Line292-301)

Line 287. Although this experiment is a simulation, I would like to know whether there have any specific

50

task/emergency observation requirements (e.g., task observation space and time). Besides, I think the ability to observe the task area is a fundamental requirement. Since the satellite flying around the earth all the time, whether a
satellite can monitor a specific area requires additional calculation. Thus, I would also like to know how those satellites and UAVs are retrieved from your database.

This experimental simulation was launched based on a mudslide disaster potential site in Shanxi Province, China, for time is not no more set, so for the monitoring coverage capability of satellite in this designated area, although it is needed to be calculated additionally, but lack of relevant conditions, we mainly considered the area coverage of UAV data here, using

60 expert experience to determine. Since the database does not now form a complete user operating system, for the retrieval of satellite and UVA data, we used an interactive approach.

In Table 3, the first indicator is "Spatial resolution of satellites", why not the UAV or the sensor? Besides, each collaborated satellite and UAV mounted the same sensor, what you would do if their mounted sensors were different?

65 Combined with the research about the emergency response process of geohazard, after the occurrence of geohazard, highresolution satellite is responsible for providing multi-temporal and multi-scale information, while UAV technology is combined with satellite data for small-scale operation, compared with the two, we believe that satellite resolution is a more important factor to be considered.

In the case of installing different types of sensors, it is necessary to establish a system of indicators according to the

70 corresponding synergistic characteristics, and the relationship mapping can be established through secondary indicators, which is our current idea, and the specific implementation process needs to be studied.

**Line 329. The mentioned Figure should be Figure 7.**

We are sorry that this was a mistake in our work and it has been corrected. (Line353)

75

90

Line 372. The result here is not clear. You claimed that the emergency response capability of good increased to 60%, but you didn't give the probability before the increase. Moreover, it seems that the third-level indexes should be sourced from satellites and UAVs, but those satellites or UAVs are not given.

This part was that we did not express clearly, has been improved in the manuscript. (Line396-403)

- 80 For the source of the third-level indexes, since we evaluate the process of the earthquake disaster emergency service system with reference to the following two papers (which are listed in the manuscript), the evaluation object is also for the emergency observation network composed by the sensors mentioned in the literature, which is a complex system and we do not have the real situation, so we use empirical simulation data here, which is easy to be misleading by simply proposing sensors. Also our work in this part is more focused on showing that the Bayesian Network evaluation network is
- 85 computationally feasible and informative. In addition, we are also accumulating more real scenario data and hope to study and improve this part as soon as possible.

 Pan, G., Tang, D.L.: Damage information derived from multi-sensor data of the Wenchuan Earthquake of May 2008. IJRS., 31(13), 3509-3519, https://doi.org/10.1080/01431161003730865, 2010.

[2] Zhang, Z., Zhang, Y., Ke, T., Guo, D.: Photogrammetry for First Response in Wenchuan Earthquake, Photogrammetric Engineering & Remote Sensing, 75(5) 510-513, https://doi.org/10.2352/J.ImagingSci.Technol.2009.53.3.030501,2009.

**Lastly, I think there lacks a criterion analysis and discussion of the experimental results. For example, although the scores of A, B, and C (section 4.1) are calculated, the reason why C is the best is not analyzed and discussed.**

A discussion and analysis of the experimental results as indicated was added to the manuscript. (Line 327-335, Line 415-418)

**Index establishment and capability evaluation of space-air-ground remote sensing cooperation in geohazard emergency response**

**Yahong Liu1, Jin Zhang2**

5

1 Department of Earth Science and Engineering, Taiyuan University of Technology, Taiyuan 030024, China;

2 Department of Surveying and Mapping Science and Technology, Taiyuan University of Technology, Taiyuan 030024, China

**Correspondence to: Jin Zhang (zjgps@163.com)**

Abstract. Geohazard emergency response is a disaster event management act that is multifactorial, time critical, task intensive and socially significant. To improve the rationalization and standardization of space-air-ground remote sensing

- 10 collaborative observations in geohazard emergency responses, this paper comprehensively analyzes the technical resources of remote sensors and emergency service systems and establishes a database of technical and service evaluation indexes using MySQL. Based on the database, we propose the method of using technique for order preference by similarity to an ideal solution (TOPSIS) and a Bayesian network to evaluate the synergistic observation effectiveness and service capability of remote sensing technology in geohazard emergency response, respectively. We demonstrate through experiments that
- 15 using this evaluation can effectively grasp the operation and task completion of remote sensing cooperative technology in geohazard emergency response. This provides a decision basis for the synergistic planning work of heterogeneous sensors in geohazard emergency response.

Keywords: Geohazard; Remote sensing cooperation; Index database; Capacity evaluation

**1** Introduction**

- 20 Geohazards are earthquakes, mountain collapses, landslides, debris flows, ground collapses, ground fissures, land subsidence and other hazards related to geological processes that endanger people's lives and property and are caused by natural factors or human activities. According to the United Nations Office for Disaster Risk Reduction (UNDRR), the human casualties caused by geological hazards since 1990 have been concentrated in the Asia-Pacific region and Africa for a long time, with 2010-2019 the decade with the highest economic losses caused by disasters (UNDRR Annual Report, 2019; UNDRR GAR
- 25 2019). To respond to sudden geological hazards and mitigate damage, it is necessary to carry out hazard emergency response quickly after the occurrence of a hazard, provide emergency assistance for victims and seek to stabilize the situation and reduce the probability of secondary damage (Johnson, 2000).

Earth observation technology provides key technical support during geohazard emergency response (Butler et al., 2005). With the development of global earth observation technology, the performance of remote sensing technology is constantly

30 improving, the number of sensors continues to increase and a multiplatform observation system for satellites, aerials,

The email address was changed

unmanned aerial systems (UASs)and the ground has gradually been established (Toth et al., 2016). There are many online resources for recording remote sensing information, and the NASA master directory (NSSDC, 2020) provides a mechanism for retrieving satellite names, classifications or launch dates to obtain descriptions of relevant satellite information and data collection. The CEOS Missions, Instruments, and The Measurements (MIM) Database is divided into Agencies, Missions,

- 35 Instruments, Measurement and Datasets modules with a focus on current and future satellites, sensors and measurement capabilities (CEOS, 2020). The Observing Systems Capability Analysis and Review tool (OSCAR, 2020) database is divided into a description of information about the satellite and its sensors and a sensor capability assessment analysis. At present, most of the Earth observation technology resources operate independently, and when faced with specific
- geohazard emergency response tasks, the space-air-ground remote sensor resources show both "many" and "few." That is, although sensor resources are abundant, it is difficult to find suitable and available sensors quickly, and this affects the efficiency of observing mission responses. The main reason is that remote sensing systems of various types are very different in terms of observation modes, applications and processing methods. In addition, resources are deployed in a distributed fashion, are described in their own independent formats, lack correlation mechanisms and cannot be detected in a timely manner (Li et al., 2012). To improve the efficiency of emergency response, a number of organizations and mechanisms have
- 45 been established internationally to synergize these resources, including the Committee on Earth Observation Satellites (CEOS), Integrated Global Observing Strategy (IGOS), International Charter Space and Major Disasters (CHARTER) and United Nations International Strategy for Disaster Reduction (UNISDR). The International Strategy for Disaster Reduction (UNISDR), United Nations Platform for Space-based Information for Disaster Management and Emergency Response (UN-SPIDER), Disaster Monitoring Constellation (DMC) and Copernicus EMS are mainly oriented to international major
- 50 disaster emergency responses such as the Wenchuan earthquake (PAN et al., 2010), Haiti earthquake (Duda et al., 2011) and Japan earthquake (Kaku et al., 2015). In addition to establishing collaborative emergency response with satellite remote sensing, in the face of the diversified needs of actual geohazard emergency response, collaboration between satellites and other multiple remote sensing platforms has become an important development direction of remote sensing technology (Li et al., 2017). This is characterized by the ability to integrate the observation advantages of each platform to effectively shorten
- 55 the observation time, expand the coverage and improve the accuracy of observation data (Asner et al., 2012; Nagai et al., 2009). Haghighi et al. (2019) used multi-SAR satellite sensors for the analysis of spatial and temporal processes of ground subsidence in the Iranian region of Drangheh, Hermle et al. (2021) used to verify the feasibility of optical remote sensing in landslide hazard warning through a combination of high-resolution satellite and UAV data and Lu et al. (2019) mapped landslide inventories based on multi-remote sensing sensor data, Ventisette et al. (2015) described data acquisition using
- 60 satellite and ground-based sensors in landslide disaster response, and Huang et al. (2017) proposed a complete set of methods for geohazard emergency investigation using UASs. In these remote sensing collaborative disaster emergency applications, by linking different types of remote sensors and coupling them to form an independent and dynamically adaptable and configurable space-air-ground remote sensing collaborative observation system, the complementary advantages of remote sensing observation platforms are brought into full play. However, there is no sensor discovery process

In the Introduction, the materials of the collaboration of heterogeneous sensors are kind of old, and it seems that some latest works are missing. Just to name a few: ...: Material updates 65 in these studies, and there is a lack of selection criteria and capability evaluation of sensors in different collaborative applications.

The observation tasks under geohazard emergencies are complex and diverse and have certain requirements in terms of timeliness and accuracy, and it is especially important for decision-makers to make comprehensive discoveries and to establish accurate collaborative planning and rapid scheduling of massive sensors in a specific emergency response situation.

- 70 How to quickly and rationally arrange the sensors that meet the geohazard emergency response needs in the sensor web environment to optimize resource utilization is the key issue in remote sensing collaborative observation. This work focuses on establishing a link between geohazard emergency response events and sensors, constructing indicators for evaluating the technical capabilities of sensors and evaluating geohazard emergency service capabilities. Wang Wei et al. (2013) proposed a mission-oriented assessment of the observational capabilities of imaging satellite sensor applications with the horizontal
- resolution, revisit period and observation error as indicators. Hong Fan et al. (2015) proposed a sensor capability 75 representation model to describe typical remote sensor capabilities for soil moisture detection applications. Zhang Siyue et al. (2019) proposed a model for evaluating the effectiveness of observations and data downlinks for low-orbiting satellites. Hu et al. (2019) constructed the observation capability information association model (OCIAM) for the selection of sensors and their combinations, and further proposed the sensor observation capability object field (SOCO-Field) to construct sensor
- 80 associations for a specific emergent geographical environment observation task (GeoTask) (Hu, 2020), and Wang et al. (2020) introduced the space-ground maximal coverage model with multiple parameters (SGMC-MP) to complete sensor mission planning. The current research data on remote sensor capabilities are relatively scarce and focus on evaluating the inherent capabilities of individual satellite remote sensors with a single object of evaluation, making it difficult to meet the needs of multisensor and multigeohazard emergency response tasks. Thus, it is necessary and timely to establish 85 collaborative observation capability indexes for space-air-ground remote sensor resources and to conduct evaluations of
- geohazard emergency response service capabilities.

**2 Data**

Given the richness of remote sensor technology resources, the service system for emergency response to geohazards has been improved in application practice. An important question how to fully discover and use the existing sensor technology to 90 meet the target observation needs and achieve the optimal effect of resource utilization for different application services. To allocate sensor resources scientifically, improve the rationality and effectiveness of cooperative observation and obtain the required information to a greater extent, this section establishes an index database for comprehensive analysis of the technical performance and emergency service system of space-air-ground remote sensing, and realizes the integrated management of the technical performance data of various types of remote sensors and emergency service information.

In the Introduction, the materials of the collaboration of heterogeneous sensors are kind of old, and it seems that some latest works are missing. Just to name a few: ...: Material updates

**95 2.1 Sensor technology resource emergency service system**

Current remote sensors can be divided into satellite, aerial and terrestrial types according to the platforms on which they are mounted (Grün, 2008). Satellite remote sensing is divided into land satellites, meteorological satellites and ocean satellites according to their fields of operation. Land satellites are mainly used to detect the resources and environment on the earth's surface and contain a variety of sensor types such as panchromatic, multispectral, hyperspectral, infrared, synthetic aperture

- 100 radar, video and luminescence (Belward et al., 2015). Meteorological satellites observe the earth and its atmosphere, and their operations can be divided into Sun-synchronous polar orbit and geosynchronous orbit (NSMC, 2020; Wang et al., 2018). Oceanic satellites are dedicated satellites that detect oceanic elements and the marine environment with optical payloads generally including watercolor water thermometers and coastal zone imagers and microwave payloads including scatterometers, radiometers, altimeters and SAR (Fu et al., 2019). The countries and regions in the world that currently have
- 105 autonomous remote sensing satellites include the United States, France, ESA, Germany, Israel, Canada, Russia, China, Japan, Korea and India. The main satellite launches are shown in Table A1. Aerial remote sensing is a technology that uses aircraft, airships and UVAs as sensor carriers for detection (Colomina et al., 2014). Different airborne remote sensing devices have been developed to face various remote sensing tasks. These devices include digital aerial cameras, LiDAR, digital cameras, imaging spectrometers, infrared sensors and min SAR. Ground remote sensing systems have two states: mobile and static. A
- 110 mobile measurement system executes rapid movement measurement by means of vehicles (e.g., cars and boats) and consist of sensors such as CCD cameras, cameras, laser scanners, GPS and inertial navigation systems (INSs) (Li et al., 2015). These can acquire the geospatial position of the target while collecting realistic images of the features. Static state measurement refers to the installation of sensors in a fixed place and includes laser scanners, cameras, ground-based SAR and surveying robots. These can form a ground sensor web through computer network communication and geographic information service
- 115 technology.

In the face of geohazard emergency responses, space-air-ground remote sensors establish associations through collaborative planning to form a collaborative observation service system based on the process of "observation-transmission-processing-distribution," as shown in Fig. 1. In the event of a geological disaster, the emergency command center responds quickly, planning observation missions according to observation needs and the current technical environment((1,2)). After remote

120 sensing systems carry out observation missions(③), the data is received, processed and distributed through the data center, providing emergency services mainly based on geographic information(④,⑤,⑥).

In Figure 1, I think it is better to number all lines but not just "observation-transmission-process-distribution", which could provide the reader a better understanding of the workflow. Modify the Fig.1

**Figure 1. Collaborative remote sensing observation service system for geohazard emergency response**

The geographic information services provided by the remote sensing emergency service system are shown in Fig. 2. These services include data processing, data products, data services, model services, functional services and warning services. Data processing refers to the process and method of obtaining effective emergency information from the collected data and includes the data processing method, feature extraction, image classification and image analysis. Data products refer to the quality and current potential of various types of remote sensing products. Data services provide disaster-related basic data, thematic data and analysis data through Web Map Service (WMS), Web Feature Service (WFS), Web Coverage Service

130 (WCS) and Web Map Tile Service (WMTS). Functional services provide quantitative, qualitative, characterization and visualization of geospatial phenomena through spatial analysis services, terrain analysis services and visualization services. Model services provide various models for calculation, analysis, anomaly identification, damage assessment, situational assessment, evaluation, decision-making and optimization. Warning services provide early warning of disasters with regard to space, time and situation. In Figure 1, I think it is better to number all lines but not just "observation-transmission-process-distribution", which could provide the reader a better understanding of the workflow. Modify the Fig.1

Line 123. "WMP" should be "WMS"; line 125. "...visualization studies...", it seems that the "studies" used here is a misspelling. Revision